

# Significant methane undersaturation during austral summer in the Ross Sea (Southern Ocean)

Wangwang Ye[1], Hermann W. Bange[2], Damian L. Arévalo-Martínez[2], Hailun He[3], Yuhong Li[1], Jianwen Wen[1], Jiexia Zhang[1], Jian Liu[1], Man Wu[1], Liyang Zhan[1]

5 1 Key Laboratory of Global Change and Marine-Atmospheric Chemistry, Third Institute of Oceanography, Ministry of Natural Resources, 361005 Xiamen, China.

2 GEOMAR Helmholtz Centre for Ocean Research Kiel, Düsternbrooker Weg 20, 24105 Kiel, Germany

State Key Laboratory of Satellite Ocean Environment Dynamics, Second Institute of Oceanography, Ministry of Natural Resources, 310012 Hangzhou, China

Corresponding to: Liyang Zhan (zhanliyang@tio.org.cn)

**Abstract.** Dissolved methane ($CH_4$) was measured at 9 stations along a transect at 75°S in the Ross Sea during austral summer in January 2020. $CH_4$ undersaturation (mean: 82±20%) was found in the water column, with a mean air-sea $CH_4$ flux density of -0.58±0.48 $\mu mol\ m^{-2}\ day^{-1}$, which suggests that the Ross Sea was a net sink for atmospheric $CH_4$ during

the austral summer. Simple box-model calculations revealed that the $CH_4$ depletion should occur in the surface mixed layer because of $CH_4$ oxidation and advection of $CH_4$-poor waters. We propose that freshwater injection caused by sea-ice melting in summer dilutes $CH_4$ concentrations within the surface layer and thus increases its potential for atmospheric $CH_4$ uptake in the Ross Sea. Thus, we argue that both $CH_4$ consumption and sea-ice melting are important drivers of $CH_4$ undersaturation, which implies that the high-latitude area of the Southern Ocean is a sink for atmospheric $CH_4$. We estimated that the

Southern Ocean (>65°S) takes up about 0.02% of the global $CH_4$ emissions and thus represents a minor sink for atmospheric $CH_4$.

## 1 Introduction

Methane ($CH_4$) is one of the most important greenhouse gases. The dry mole fractions of $CH_4$ in the atmosphere have

increased continuously since the onset of the industrial revolution, contributing more than 20% of the anthropogenic radiative forcing in the lower atmosphere (*IPCC*, 2021; *Saunois et al.*, 2020). In addition, the ice-covered parts of the Antarctic are now known to be a reservoir of organic carbon (*Priscu et al.*, 2008) which might fuel $CH_4$ production, suggesting that the Antarctic ice sheet may be a neglected but important component of the global $CH_4$ budget (*Wadham et al.*, 2012). However, the very limited number of direct observations of $CH_4$ in the water column of the Southern Ocean





constrains our ability to assess oceanic CH$_4$ dynamics and emissions. Hence, more field data should be collected to decipher
the role of the Southern Ocean in global CH$_4$ cycling.

The Ross Sea (Figure 1a), located along the Antarctic margin between Cape Adare and Cape Colbeck, is one of the
largest consistently forming polynyas in the Southern Ocean (*Arrigo and Dijken*, 2004). The shelf is usually free of sea ice
during summertime in the Antarctic (December to February). During winter, winds blow northward along the Ross Ice Shelf

and form discrete coastal polynyas and areas of exposed surface water surrounded by dense multiyear ice (*Orsi and
Wiederwohl*, 2009). Evidence for the presence of gas hydrates in the western Ross Sea was inferred from a bottom-
simulating reflector, indicating a potential source of CH$_4$ under the seabed (*Geletti and Busetti*, 2011). However, only a few
studies have addressed the distribution of CH$_4$ in the water column of the Southern Ocean, and these studies have rarely
focused on the Ross Sea. In the 1970s, surface CH$_4$ concentrations were investigated at several locations in the vicinity of the

Ross Sea, showing a sink for atmospheric CH$_4$ that was explained by the microbial consumption of CH$_4$ (*Lamontagne et al.*,
1974). The CH$_4$ distribution around the Antarctic Peninsula and in the Weddell Sea was investigated in the 1990s and 2000s,
suggesting a net uptake of CH$_4$ in open ocean water, which was controlled by mixing between surface water and deep water
in which microbial consumption decreased the CH$_4$ content (*Heeschen et al.*, 2004; *Tilbrook and Karl*, 1994). Moreover,
CH$_4$ undersaturation was observed in high-latitude areas (>65°S) of the Southern Ocean, except for the circumpolar areas

(Pacific sector of the Southern Ocean) where the surface water showed a minor source of atmospheric CH$_4$ due to local CH$_4$
release in the water column (*Yoshida et al.*, 2011; *Bui et al.*, 2018).

Sea ice can be a physical obstruction for air-sea exchange of CH$_4$, and the storage of CH$_4$ in ice is usually a minor
source compared to brine and under-ice seawater (*Crabeck et al.*, 2014; *Zhou et al.*, 2014). Thus, areas such as the Arctic
Ocean have been found to produce short-term pulses of CH$_4$ emissions during the melting season (*Thornton et al.*, 2016). In

contrast, the CH$_4$-depleted regions of the Southern Ocean, such as the Ross Sea, may act as CH$_4$ sinks after freshwater
injection due to ice melting, which is similar to the uptake of atmospheric CO$_2$ in the ice-free polar seas (*Rysgaard et al.*,
2011; *Sandrini et al.*, 2007). However, more observations are needed to understand the CH$_4$ variability in the Southern
Ocean. Here, on the basis of our results, we propose (i) that the mixing of water masses controls the CH$_4$ distribution in the
Ross Sea and (ii) that sea-ice melting enhances the ability of surface water to take up atmospheric CH$_4$ during summertime.

**2 Materials and Methods**

**2.1 Hydrographic conditions and water mass classification**

Five water masses were identified in the Ross Sea according to their temperature/salinity properties (Figure 2a; Table
S1) and the definitions used in previous studies (*Narayanan et al.*, 2019; *Orsi and Wiederwohl*, 2009; *Schodlok et al.*, 2016;
*Zoccarato et al.*, 2016). Antarctic Surface Water (AASW) carried by coastal currents (blue arrow in Figure 1a) flows

westward along the outer ice shelf, occupying the surface layer (<100 m) of the water column. It is a cold, fresh, and oxygen-
rich layer due to melting sea ice at the surface and sufficient contact with the atmosphere. Shelf Water (SW) is formed





during winter when brine rejection occurs due to sea-ice formation, causing the local surface water to become dense and sink to the bottom along the slope. A thermohaline mechanism drives the overturning circulation of SW in the ice cavity; therefore, SW experiences extra cooling from the ice cavity and becomes Ice Shelf Water (ISW) when flowing off the ice

sheet (green arrow in Figure 1a), which has a below-freezing point temperature (*Lewis and Perkin*, 1986). Warmer, saltier, and oxygen-poor Circumpolar Deep Water (CDW) carried by the Antarctic Circumpolar Current intrudes into the edge of the Ross Sea. Afterward, the Antarctic Slope Current (red arrow in Figure 1a) brings it to be mixed with AASW, which then reaches farther south and becomes Modified Circumpolar Deep Water (MCDW) (red dashed arrow in Figure 1a) (*Orsi and Wiederwohl*, 2009). In addition, Modified Shelf Water (MSW) is formed when near-freezing SW is vertically mixed with

relatively warm MCDW, which is the classic mechanism to produce Antarctic Bottom Water (*Assmann and Timmermann*, 2005).

The relative abundance of a specific water mass at each station was estimated by the percentage of local volume following the description by *Orsi and Wiederwohl* (2009). Briefly, the spatial distribution of the potential temperature (θ)-salinity diagram provides the thickness of a certain water mass. Assuming all stations had the same sectional area, the

volume of each water mass at a specific station was simply determined by the depth at which the water mass was encountered (Table S2). The volumetric percentages of each water mass (Table S3) were then calculated from the total volume divided by the volume of a specific water mass at a specific station.

**2.2 Sampling and analysis**

Seawater samples for $CH_4$ analysis were collected onboard R/V "Xuelong 2" during the 36[th] Chinese Antarctic

Research Expedition (CHINARE) to the Ross Sea from 3–6 January 2020. The $CH_4$ distribution was measured at nine out of ten stations on a transect along 75°S from 164°E−182°E. The samples were collected with a rosette water sampler equipped with Niskin bottles, following the sampling strategies described by *Zhan et al.* (2018). Briefly, seawater was transferred to 250 mL borosilicate glass bottles with standard taper stoppers (Corning PYREX®, USA), which were sealed with Apiezon grease (Sigma-Aldrich®, USA) and stored in the dark at 4 °C after 180 µL of saturated $HgCl_2$(aq) solution was added.

Samples were analyzed immediately after shipment to the home laboratory (storage time < six months). Hydrographic data were collected with a SBE-911plus conductivity-temperature-depth (CTD) unit (Sea-Bird, USA) that measured the salinity (±0.002 psu), temperature (±0.001 °C), and pressure (0.015% of full-scale range). Triplicate or duplicate subsamples were dispensed to 20 mL glass vials prior to analysis. $CH_4$ was extracted from the subsamples by a purge-and-trap method (Text S1; Table S4–S5; Figure S1), and its dry mole fraction (water vapor was removed by a Nafion™ dryer) was determined with

a gas chromatograph (Agilent 7890A) equipped with a flame ionization detector (Zhang *et al.*, 2004). $CH_4$ saturation (Sat, %) in seawater is given by the following equation:

$$\text{Sat} = {C_{obs}}/{C_{eq}} \cdot 100\% \tag{1}$$





where $C_{obs}$ is the observed concentration of $CH_4$ in seawater and $C_{eq}$ is the concentration in equilibrium with the

atmosphere calculated from the solubility coefficient (*Wiesenburg and Guinasso*, 1979) with the *in-situ* temperature (-

1.9−1.4 °C) and salinity (33.68−34.84) at the time of sampling and the atmospheric $CH_4$ dry mole fraction of 1.82 ppm
(monthly mean during the sampling period at the South Pole Observatory, available from NOAA Global Monitoring
Laboratory, *Dlugokencky et al.*, 2020). Using a contemporaneous atmospheric $CH_4$ mole fraction for all water depths led to
an underestimation of less than 1% for the equilibrium concentration of dissolved $CH_4$ in view of the residence time of about
4 years of the Ross Sea shelf water (*Trumbore et al.*, 1991). The overall mean analytical error of $C_{obs}$ was ±0.3 nmol L$^{-1}$,

with a corresponding mean error of ±10% for the $CH_4$ saturations.

## 2.3 Sampling and analysis

The flux density ($F_{ase}$, µmol m$^{-2}$ day$^{-1}$) of $CH_4$ across the air-sea interface was estimated by the following equation:

$$F_{ase} = k_w \cdot (C_{obs} - C_{eq})  \tag{2}$$

where $k_w$ is the gas transfer coefficient (m day$^{-1}$) and $C_{obs}$ is the measurement from the uppermost sampling depth (2−5

m). We used an empirical model to determine $k_w$, which is a function of the molecular diffusivity of $CH_4$ in water (*Jähne et al.*, 1987), the kinematic viscosity of water, and the wind speed (*Wanninkhof*, 2014). The molecular diffusivity and
kinematic viscosity, in turn, were computed as functions of the temperature and salinity. We used the average of the daily
wind speeds over the Ross Sea during the sampling period (5.8±0.1 m s$^{-1}$) which was obtained from the National Centers for
Environmental Prediction (NCEP) and the National Center for Atmospheric Research (NCAR) reanalysis data

(http://www.psl.noaa.gov/data/gridded/data.ncep.reanalysis.derived.html). This regional mean wind speed was in the same
range (5.2−6.3 m s$^{-1}$) as that reported from the Weddell Sea (*Tilbrook and Karl*, 1994) but lower than the 30-year January
average (6.8 m s$^{-1}$) from the Southern Ocean south of 50°S (*Bui et al.*, 2018).

## 2.4 CH$_4$ budget in the mixed layer

In order to roughly estimate the $CH_4$ production/consumption in the mixed layer, we performed a mass balance

calculation of $CH_4$ fluxes into and out of the surface mixed layer. The mixed layer was defined as the water depth where a
potential density difference of 0.125 kg m$^{-3}$ and a temperature difference of 0.5 °C in comparison with the ocean surface was
computed (*de Boyer Montégut* et al., 2004). The $CH_4$  fluxes in the surface mixed layer should be balanced and can be
expressed by the following equation:

$$-F_{ase} + F_{vd} + F_{ad} + F_x = 0  \tag{3}$$

where $F_{ase}$ (µmol m$^{-2}$ day$^{-1}$) is the air-sea exchange of $CH_4$ across the ocean/atmosphere interface obtained from
equation (2). The $F_{vd}$ (µmol m$^{-2}$ day$^{-1}$) is the vertical (diapycnal) diffusion of $CH_4$, which can be obtained from Fick's First

law: $F_{vd} = K_z \cdot {}^{d_C}/_{d_h}$, $h$ is depth (m), $d_c/d_h$ is the vertical $CH_4$ gradient measured at each station, $K_z$ (m$^2$ s$^{-1}$) is the diapycnal

diffusivity. We used a K$_z$ of 10$^{-4}$ m$^2$ s$^{-1}$ which was suggested to be representative for the Southern Ocean (*Mashayek* et al.,





2017). $F_{ad}$ (μmol m$^{-2}$ day$^{-1}$) is the lateral CH$_4$ transport (advection) into/out of the mixed layer which we assumed in a first

approximation to be negligible. $F_x$ (μmol m$^{-2}$ day$^{-1}$) is the unknown in-situ source/sink (such as production/oxidation) in the mixed layer, which was estimated by the balance of other flux terms. Positive values represent a CH$_4$ source in the mixed layer while negative values represent a CH$_4$ sink in the mixed layer.

## 3 Results

### 3.1 Section distribution of temperature, salinity, and CH$_4$

The water column was stratified within the upper 50 m, with density differences (△σ) of 0.185−0.522 kg m$^{-3}$ among all stations. Other than the influence of wind, temperature differences (0.04 °C m$^{-1}$) contributed partly to the stratification of waters in the western part of the study area (R1−R3) (Figure 1b). In contrast, the westward intrusion of warm AASW could have partly contributed to the water stratification in the middle and eastern region of the study area. For example, an increase in temperature (0.01 °C m$^{-1}$, 0–25 m) contributed the most to the stratified water at station R5 and the halocline (0.01 psu m$^{-1}$,

0–25 m) probably drove the strong stratification at the stations R9. Below 50 m, uniform water (<0.0001 °C m$^{-1}$ and <0.0002 psu m$^{-1}$) was found over the west of Mawson Bank with slightly heavier water in the trough. Lateral transport of water masses by the slope current occurred in the edge of the Ross Sea, where a southward intrusion of saline MCDW was observed during our cruise (a decrease in MCDW contribution from station R10 (300 m) to R7 (100 m), Figure 2b). Consequently, the water column was vertically classified into three layers (AASW, SW/MSW, and MCDW) according to

their hydrographic features.

Generally, the seawater was undersaturated in CH$_4$ (mean ± std: 82±20%), with a relatively high mean CH$_4$ saturation (88±19%) near the coast (stations R1−R5) compared with that in the offshore areas (74±19%, stations R6−R10). In the west, waters in the upper 50 m had the highest CH$_4$ saturation, with a mean of 95±20%. Below 50 m, CH$_4$ saturation decreased to 86±18% and 84±12% at depths of 50−400 m and below 400 m, respectively. In the east, station R10 (where AASW

dominated the water column) had the lowest CH$_4$ saturation (63±8%) compared to those at the other stations. The highest CH$_4$ saturation (147%) was found at station R7 at 200 m together with an anomalous warm-water mass (-1.0 °C compared with an ambient temperature of -1.7 °C) at the same depth.

### 3.2 Water mass distribution

SW was the dominant water mass among the study areas west of 175°E, accounting for 60−90% of the total water

volume (Figure 2b). It formed near Victoria Land, and its contribution decreased to the east. With its high salinity, SW would be expected to be trapped in deep troughs (>400 m) and not flow out to the Ross Sea Basin (<2% east of the Joides Trough). In contrast, MSW increased from ~5% in the west to 20−50% in the east. Correspondingly, AASW and MCDW were distributed in the eastern part of the shelf, with the largest proportions of AASW and MCDW at station R10 (65% for AASW and 34% for MCDW) and the lowest proportions at stations R1−R3 (0−6% for AASW and 0−5% for MCDW).



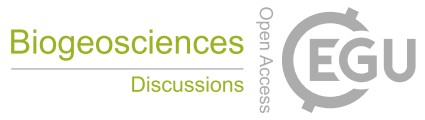

Station R9 was likely located in the most heterogeneity region, which consisted of near equal contributions of AASW, MSW, and MCDW. A small fraction of ISW (10−19%) was found only near the ice sheet, where supercooled water could be formed in the cavity below the ice sheet.

### 3.3 Sea ice distribution and surface CH₄

During the austral summer of 2019/2020, the sea ice in the Ross Sea began to melt in December near the western coast
of the Ross Ice Shelf (Figure 3a). Afterward, the ice-covered areas decreased rapidly in January, and the western part of the Ross Sea became an ice-free area in February. We found that during our sampling period (January 2020), the Pennell Bank (stations R6−R10) had become an ice-free area (which may have already endured a 30-day ice-free period by then, Figure 3), while sea ice was still present at the Mawson Bank. Lower CH₄ saturation at the surface (76±13%) was found at the Pennell Bank while near-equilibrium saturation (91±8%) was found at the Mawson Bank (stations R1−R5, ice-covered area:
165 50±30%).

Different endpoints were chosen to quantify the factors that controlled CH₄ undersaturation (Figure 3e). Considering the water stratification and relatively slow gas exchange with the atmosphere (4 yr residence time for SW, *Trumbore et al*., 1991), we assumed that the CH₄ saturation of 84% (mean saturations as derived from our measurements below 400 m water depth) in deep waters (>400 m) could represent the endpoint in the SW in early summer. In addition, the sediment-
influenced water at station R7 was assumed to be a CH₄ source. We found that mixing among AASW, SW, and CDW was responsible for the CH₄ undersaturation in the shelf area. A total decrease of 50% ice-covered area drove a decrease in surface CH₄ saturation by an average of 22% (from 91±8% to 76±13%).

### 3.4 Box-model calculation

The surface mixed layer was 25 m at stations R1−R6 (in the west) and 50 m at R7−R10 (in the east) and therefore we
calculated two box models for comparison. At stations R1-R6, the air-sea exchange accounts for a mean uptake of 0.37 µmol m⁻² day⁻¹ from the atmosphere, the mean vertical diffusion rates ranged from -0.21 to 0.62 µmol m⁻² day⁻¹, with a mean value of 0.17 µmol m⁻² day⁻¹. If we assume that the lateral transport of CH₄ is zero, the $F_x$ was estimated as -0.54 µmol m⁻² day⁻¹. For stations R7-R10, because we only have vertical profile CH₄ data at stations R9 and R10, we calculated the $F_x$ to be -0.83 µmol m⁻² day⁻¹ for station R9 and -1.52 µmol m⁻² day⁻¹ for station R10, respectively, which was 2-3 times higher than that at
stations R1−R6.

## 4 Discussion

### 4.1 Role of water mass mixing on CH₄ distribution

The fate of CH₄ in the Ross Sea mainly results from mixing, air-sea exchange, and microbial consumption. Previous studies have indicated that microbial oxidation of CH₄ during deep water transport results in CH₄ undersaturation and causes



CH$_4$-depleted surface waters after an upwelling of the deep water in the Southern Ocean (*Tilbrook and Karl*, 1994; *Heeschen et al.*, 2004). Hence, the CH$_4$-poor CDW may play important role in contributing to the lowest CH$_4$ saturation at the edge of the Ross Sea. MCDW, which is fed by CDW, contributes to CH$_4$ undersaturation by mixing with MSW (Figure 3e). In addition, AASW, entrained by the Coastal Current (Figure 1a) and flowing progressively westward along the Ross Sea shelf, results in a decreasing CH$_4$ gradient from west to east in the upper 200 m (Figure 2c). Within these depths, negative (positive)

correlations between CH$_4$ saturation and the percentage of AASW (SW+MSW) are found at all stations (Figure S2). Hence, after the effect of microbial oxidation of CH$_4$, mixing between different water masses determines the distribution of CH$_4$ undersaturation in the Ross Sea. In contrast, the oversaturation of CH$_4$ (106±5%) found in this study, including the near-equilibrium saturation (e.g., stations R1−R5), may originate from surface water that is sufficiently exposed to the air in summer and subsequently concentrated and transported due to ice formation in winter.

At the surface (<100 m), AASW is the main source for water undersaturated in CH$_4$ in the Ross Sea, which is probably caused by dilution with melted sea ice. During austral winter (June−August), sea-ice formation in the Ross Sea converts local upper waters (i.e., AASW and MCDW) into SW with a high density, which in turn sinks along the continental slope and produces a well-mixed water column (*Oris and Wiederwohl*, 2009). Hence, the dissolved CH$_4$ in surface water during winter would be transported to deep water and involved in long-term cycles that would be modified by mixing, oxidation,

and supplementation from sedimentary sources. When sea ice melts in summer, seawater with undersaturated CH$_4$ concentrations then continues to be diluted by melting freshwater, which in turn leads to a continuous decrease in the CH$_4$ saturation within the surface layers; this situation is counteracted by the uptake of CH$_4$ from the overlying atmosphere. For example, we found that the CH$_4$ saturation was decreased by 22% at the Pennell Bank (due to the complete melting of sea ice) compared to that at the Mawson Bank (due to slow changes in sea-ice cover and uptake of CH$_4$ from the atmosphere) in the

Ross Sea (Figure 3d). As the ice-free areas increase, CH$_4$ saturation changes dynamically in the seawater due to mixing between water masses and/or exchange with the atmosphere (Figure 3e). Thus, the magnitude of sea-ice melting may determine the degree of CH$_4$ saturation in the surface water and strengthen the capability of the Ross Sea to take up CH$_4$ from the atmosphere.

### 4.2 CH$_4$ removal in the mixed layer

The surface mixed layer plays an important role in connecting the atmosphere and the deep ocean, determining the CH$_4$ cycling in the Ross Sea. Our box-model results indicate that the changes in the CH$_4$ concentrations of the mixed layer were influenced by air-sea exchange (40−70%) in the west and increased its contribution (>90%) in the east. In contrast, the vertical diffusion of CH$_4$ was a source (upward flux) to the mixed layer in the west (except for station R1) but changed to a sink (downward flux) in the east. The sea ice distribution may responsible for the differences in the contribution of $F_{ase}$ and

$F_{vd}$. In the east, the sea ice was melting or incompletely melting during our sampling period. The physical mixing between surface water and injected freshwater causes (1) a rapid decline in CH$_4$ concentration at the surface and thus resulting in an uptake of CH$_4$ from the atmosphere and (2) leads vertical diffusion of CH$_4$ from deeper waters into the mixed layer. In the





east, after completely melting of sea ice and expose to the air for a 30-day ice-free period (Sect. 3.3), $CH_4$ concentrations in the mixed layer exceeded the $CH_4$ concentrations in deeper waters due to continuously uptake of $CH_4$ from the atmosphere

and thus $CH_4$ began to diffuse into the deeper waters.

The continuous uptake of $CH_4$ in the surface water (from the initiation of sea-ice melting to a month after the ice-free period) suggests the presence of $CH_4$ removal process(es) in the mixed layer, which is reflected by the negative values of $F_x$ calculated from our box model. A previous study reported a microbial oxidation rate of $CH_4$ for the Weddell Sea (Southern Ocean) to range from 0.34 to 1.03 µmol m$^{-2}$ day$^{-1}$, which is comparable to our results. Hence, we argue that $CH_4$ oxidation is

an important driver to reinforce the $CH_4$ undersaturation in the Ross Sea. In addition, the $CH_4$-consumption in the mixed layer were 2-3 times higher in the east than that in the west. Recalling that the contribution of the CDW was decreased from the east to the west (Figure 2b) with a relatively low $CH_4$ (Figure 3e), we would expect a stronger advection of $CH_4$-poor waters in the east may contribute to the higher rate of $CH_4$ depletion. Though the $CH_4$ removal rates were roughly estimated, the compared results among stations showed some regulations and suggest the presence of considerable $CH_4$ consumption in

the mixed layer, which is the prerequisite of atmospheric $CH_4$ uptake in the Ross Sea.

### 4.3 $CH_4$ uptake in the Southern Ocean

Unlike most oceanic areas, where supersaturation of $CH_4$ is found in the surface water, the Southern Ocean is characterized by significant undersaturation of $CH_4$ in the surface layer, which will result in a net take up of $CH_4$ from the atmosphere, similar to the case of carbon dioxide in the Ross Sea (*Rysgaard et al.*, 2011; *Sandrini et al.*, 2007). Almost all

previous studies of $CH_4$ in the Southern Ocean have shown different degrees of $CH_4$ undersaturation in the surface layer (Table 1), except for the circumpolar area near the Adélie Coast, which is influenced by local $CH_4$ production (*Yoshida et al.*, 2011). We calculated a mean $CH_4$ flux density of -0.54±0.48 µmol m$^{-2}$ day$^{-1}$ (with a range from -1.56 to -0.13 µmol m$^{-2}$ day$^{-1}$), which indicates that the Ross Sea was a sink for atmospheric $CH_4$ at the time of sampling, in line with previous studies (Table 1). Despite the increase in the dry mole fractions of atmospheric $CH_4$ during the past decades, our results indeed show

that the high-latitude regions of the Southern Ocean still maintain the potential to take up atmospheric $CH_4$.

The overall $CH_4$ uptake by the Ross Sea (extrapolated to a surface area of 9.6×10$^5$ km$^2$) was 0.001 Tg $CH_4$ during the austral summer (December−February). Thus, the Ross Sea counteracted 0.01–0.02% of the global oceanic $CH_4$ release of 6–12 Tg $CH_4$ yr$^{-1}$ (*Weber et al.*, 2019). Extrapolating this result to the high-latitude region (>65°S) of the Southern Ocean (Table 1), which accounts for ~6% of the global ocean surface area

(www.ngdc.noaa.gov/mgg/global/etopo1_ocean_volumes.html), a total of 0.02 Tg $CH_4$ (accounting for 0.2−0.3% of the oceanic $CH_4$ release) was calculated to be taken up during the three ice-free months by the Southern Ocean surface seawater after sea-ice melting in summer (Figure 3a−c). Notably, however, the variability in wind speed could substantially affect this very rough estimate. For example, it is suggested that moderate winds may contribute to the slow equilibration, where surface seawater is undersaturated with $CH_4$ (*Heeschen et al.*, 2004). Here, we use the daily average wind speed (NCEP

product), which has previously been shown to be underestimated (*Borges et al.*, 2018), possibly resulting in a conservative

estimate of the CH$_4$ flux densities. Moreover, the Ross Sea was covered by a large amount of sea ice in January 2020 (Figure 3b). Therefore, if 50% sea-ice melting would decrease CH$_4$ saturation by 22%, we would expect an increase of 44% in the CH$_4$ flux density (from the atmosphere to surface water) after the total disappearance of sea ice. This means that high-latitude regions are capable of take up atmospheric CH$_4$ during periods of sea-ice decline and could enhance CH$_4$ uptake in

the Southern Ocean when sea-ice retreats even farther in the future (Figure S3). However, more studies are needed to understand the extent and variability of the CH$_4$ uptake and how ongoing human activities affect future uptake.

## 5 Conclusions

Our measurements in January 2020 show significant undersaturation of CH$_4$ in the water column during austral summer in the Ross Sea. We find that the CH$_4$ oxidation may be the prerequisite of CH$_4$ undersaturation and that sea-ice

melting is likely to enhance the degree of surface CH$_4$ undersaturation, which increases the capability of surface water in the Ross Sea to take up CH$_4$.

The distribution and air-sea flux of CH$_4$ has been studied in coastal seawaters but have rarely focused on the high-latitude area of the Southern Ocean. Our measurement of CH$_4$ in the water column of the Ross Sea shelf demonstrate, for the first time after 1972 (*Lamontagne et al.*, 1974), that the Ross Sea was a sink for atmospheric CH$_4$ at the time of our sampling

in January 2020, which underlines the significance of the role of the Southern Ocean for both the regional and global atmospheric CH$_4$ budget. We anticipate that ongoing studies to be conducted during summer and winter, combined with a large spatial coverage, will lead to improvements in our understanding of the underlying mechanisms and processes as well as the magnitude of the CH$_4$ undersaturation in the high-latitude regions of the global ocean.

## Appendix A

Supporting information Text S1, Fig. S1−S3, and Table S1−S5.

## Appendix B

Dataset.

## Data Availability

Data from this study are available at the National Arctic and Antarctic Data Center (NAADC) (DOI:

10.11856/SNS.D.2021.005.v0) via the link:

https://www.chinare.org.cn/en/metadata/172181e0-9bbd-4069-998e-ddeef3b6abe8

To obtain data from NAADC, a registered account is necessary due to the data management policy required by government and national funders. Please follow the instructions (https://www.chinare.org.cn/en/help/using-help) to obtain the free license and access the data. For convenience, data are also provided in the Appendix B.





The CH$_4$ flux density in Table 1 and the definition of water masses in Table S1 were extracted from published literature as indicated in the tables.

## Author Contributions

W.Y., L.Z., and Y.L. designed the study with input from other coauthors; H.H. provided the hydrographic data; W.W. and J.Z. conducted the experiments; J.L and M.W. analyzed the results. W.Y. drafted the manuscript with input from H.W.B.,

D.L.A.M., and all other authors.

## Competing interests

The authors declare that they have no conflict of interest.

## Acknowledgements

The authors wish to thank our colleagues from the Third Institute of Oceanography, MNR, China, particularly Yuanhui

Zhang and Derong Zhao, and the crew of R/V "Xuelong 2" for help with the sampling. We also thank colleagues from the Polar Research Institute of China for providing hydrographic data. This study was supported by the Scientific Research Foundation of the Third Institute of Oceanography, MNR (HaiSanKe2020004, 2019033, and 2018032), the Fujian Provincial Natural Science Foundation of China (2020J01102 and 2019J05147), the National Natural Science Foundation of China (42006040 and 41906193), and the Ministry of Natural Resources of the PRC (IRASCC2020−2022). We thank two

anonymous reviewers for their constructive comments which helped to improve the manuscript significantly.

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

**Table 1.** Summary of saturation and air-sea flux densities of $CH_4$ in the Southern Ocean (60–75°S).

| Study area | Date | Latitude | Atmospheric mol fraction, ppm | Saturation % | Flux density $\mu mol\ m^{-2}\ day^{-1}$ | References |
|---|---|---|---|---|---|---|
| Ross Sea | Dec. 1972 | 70−77 °S | 1.36 | 64 | -1.1[*] | Lamontagne *et al.* (1973) |
| Antarctic Peninsula (offshore) | Dec. 1986– Mar. 1987 | 62−64 °S | 1.51 | 87 | -0.35 | Tilbrook and Karl (1994) |
| Weddell Sea | Mar. –May 1998 | 60°S | 1.69 | 75 – 94 | -0.5 | Heeschen *et al.* (2004) |
| Adélie Coast | Dec. 2001– Feb. 2002 | 60−66 °S | 1.71 | 87 – 130 | -1.0−1.2[*] | Yoshida *et al.* (2011) |
| Indian/Pacific Ocean sectors | Dec. 2012– Feb.2013 | 60−65 °S | 1.75 | 88– 105 | -3.9−0.5 | Bui *et al.* (2018) |
| Ross Sea | Jan. 2020 | 75°S | 1.82 | 86 | -0.58 | This study |

*Values were not given but recalculated by using parameters from the specific reference. The atmospheric mixing ratio

and NCEP reanalysis wind speed taken from the sampling year were used in the flux estimation.



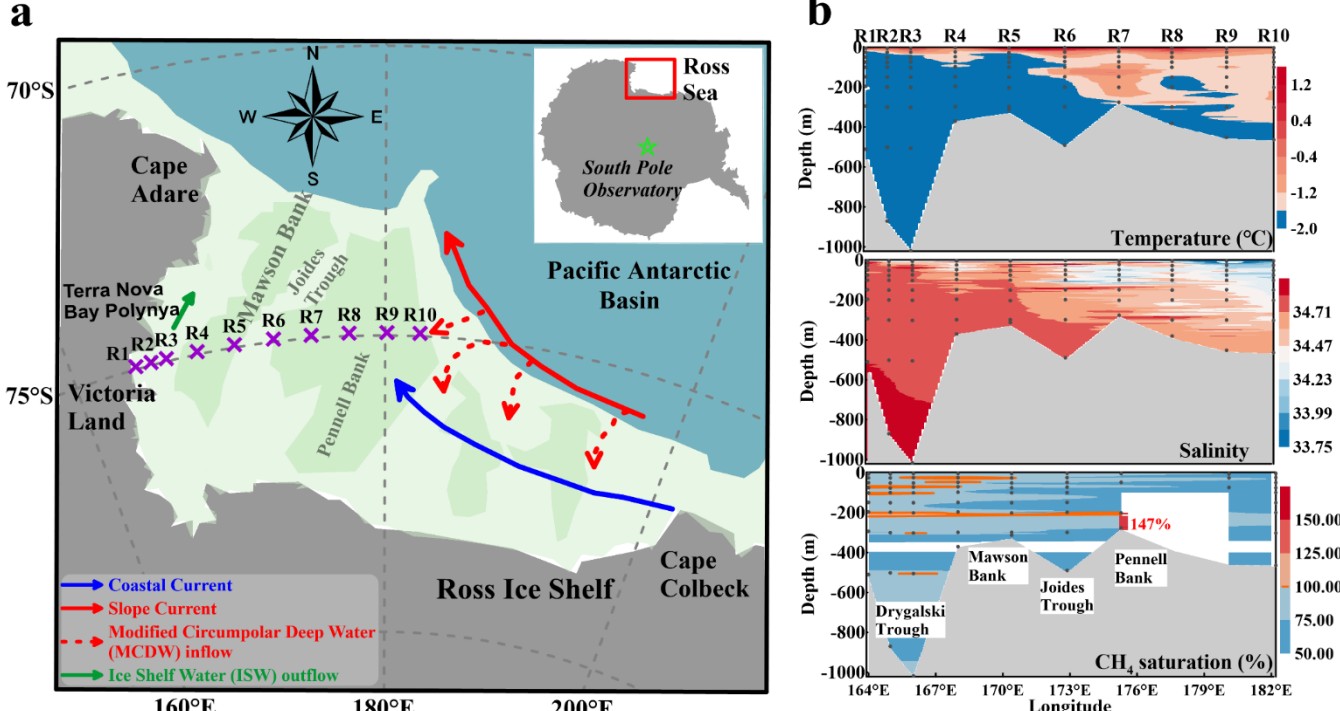

**Figure 1: Study location and distribution of sampling sites along the transect in the Ross Sea.** a) Map of the study area in the Ross Sea with an inset indicating its location in the Southern Ocean. Sampling stations are marked by purple crosses with the station number. The arrows represent schematic flows of currents during this study (*Stewart*, 2018). The light green shaded area represents the Ross Sea shelf, with a mean water depth of less than 500 m. The blue shaded area represents the outer shelf area, with a mean water depth greater than 1000 m (*Marsay et al.*, 2017; *Padman et al.*, 2009). Major banks are labeled on the map. The green star indicates the location of the South Pole Observatory. b) Vertical distributions of temperature, salinity, and $CH_4$ saturation along the transect in the Ross Sea. The contour line of 100% $CH_4$ saturation (equilibrium with the atmosphere) is shown by the orange line. The source point of $CH_4$ released from sediment is marked with a red contour at station R7 (see text). The white boxes indicate that no $CH_4$ data were available.





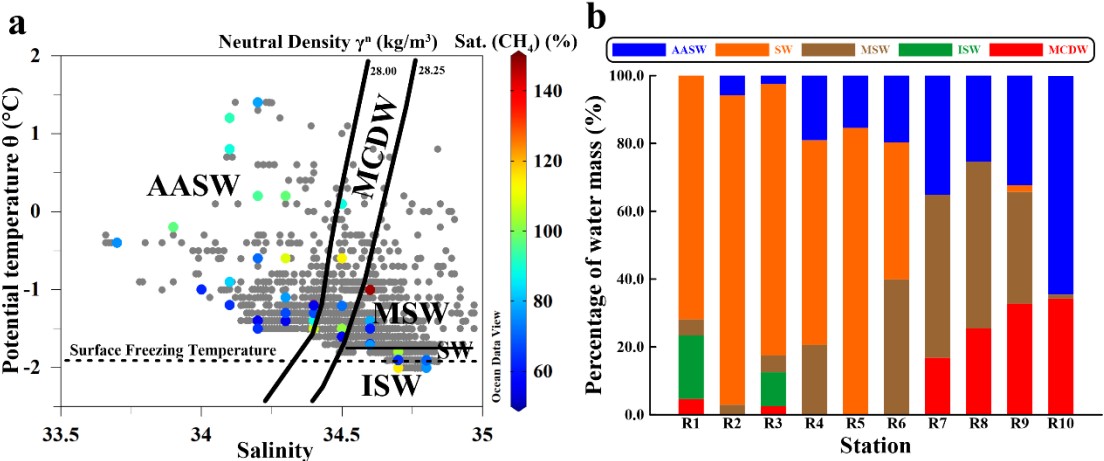

**Figure 2: Distribution of the main water masses in the Ross Sea during the austral summer of 2020.** a) Potential temperature θ-salinity scatter plot for the sampling stations. Water masses are classified as previously described (*Narayanan et al.*, 2019; *Orsi and Wiederwohl*, 2009; *Williams et al.*, 2016; *Schodlok et al.*, 2016; *Zoccarato et al.*, 2016). Solid lines show the 28.00 and 28.25 kg m$^{-3}$ neutral density $\gamma^n$ surfaces (*Jackett and Mcdougall*, 1997), which are used to define Antarctic Surface Water (AASW, lower bound) and to separate the Circumpolar Deep Water (CDW) and modified Circumpolar Deep Water (MCDW) from the Antarctic Bottom Water. The dashed horizontal line shows the surface freezing temperature of seawater. Dots are highly overlapping in the range of Shelf Water (SW). b) The proportions of water masses at each station.



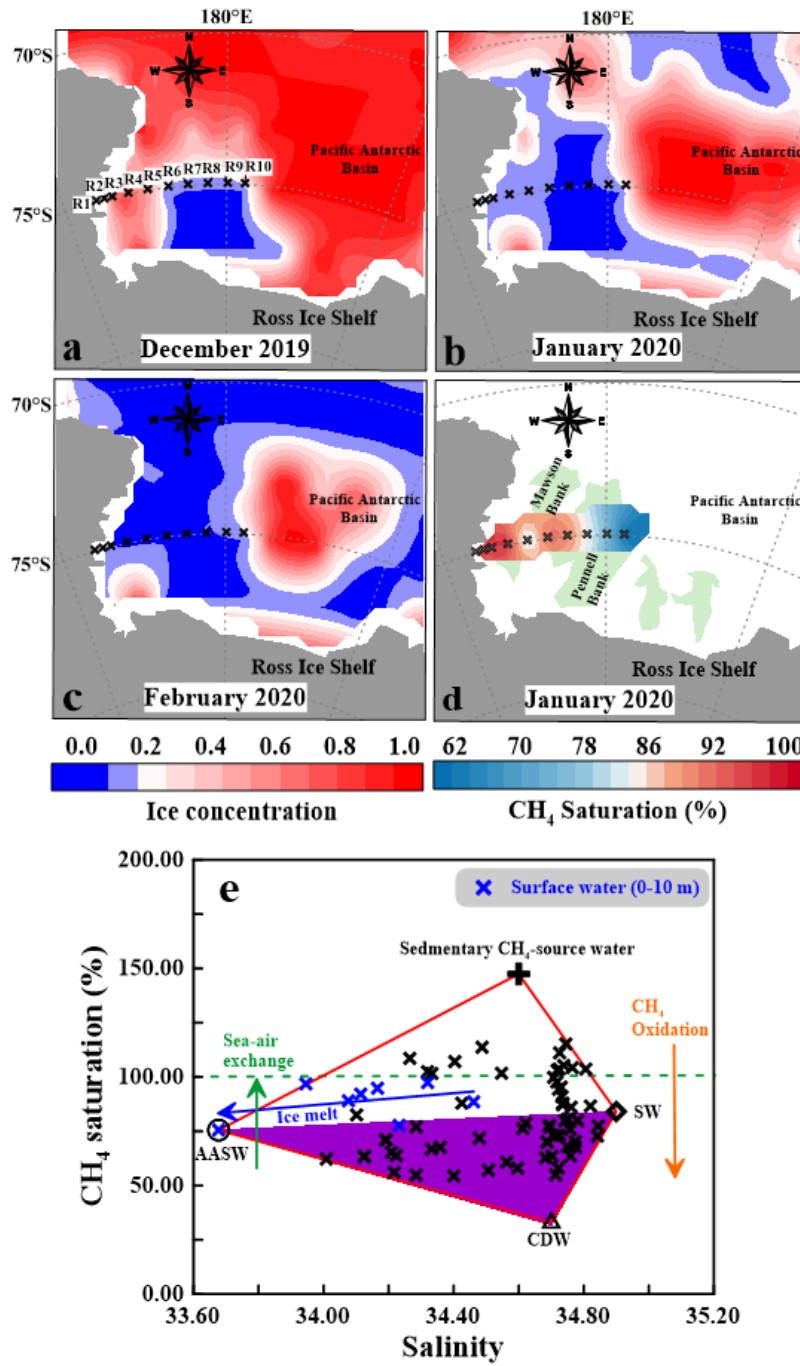

**Figure 3: Distribution of ice concentrations and the correlation between salinity and CH₄ saturation.** a-c) Changes in monthly mean ice concentrations. d) Spatial distribution of surface CH₄ saturation values in the Ross Sea during the sampling period (January 2020). Pennell and Mawson Banks are marked with light green shading. Sea ice data were obtained from the Physical Sciences Laboratory, NOAA



450 (http://www.psl.noaa.gov/data/gridded/data.ncep.reanalysis.derived.html). e) Data and model simulations of factors influencing $CH_4$ undersaturation on the Ross Sea shelf. Arrows represent controlling processes (blue for ice melt, green for air-sea exchange, orange for microbial oxidation, and red for water mixing). The green dashed line represents $CH_4$ in equilibrium with respect to the atmosphere. The open circle represents the endpoint of AASW that was obtained from station R9 with the lowest salinity; the plus represents the endpoint of sediment-influenced water that was obtained from the near-

455 bottom water (200 m) at station R7; the diamond represents the endpoint of SW, which was the mean value obtained from stations R1–R3 at depths greater than 400 m; the triangle represents the endpoint of CDW that was obtained from *Heeschen et al.* (2004). The purple shading indicates the mixing of MCDW and MSW.