# Peer review of "Significant methane undersaturation during austral summer in the Ross Sea (Southern Ocean)"

_Biogeosciences, 2021_

## Referee Comment (RC1)

Review of :

**Significant methane undersaturation during austral summer in the Ross Sea (Southern Ocean)**

Ye et al.

**General Comments**

This paper presents a nice data set on $CH_4$ water concentration profiles in the Ross Sea during the Summer 2020. This is rare data, which, per se, deserves publication. The data set is interpreted in terms of potential driving processes such as water mass mixing, sea ice melting, biological consumption/production and air-ocean exchange processes.

I have however reservations on several aspects of the paper, both in terms of structure of the paper, methodology and conclusions.

My major concern is that the paper argues that $CH_4$ profiles in the mixed layer demonstrate the impact of sea ice melting on the $CH_4$ dynamics in the area. First of all, I have doubts on the pertinent use of $CH_4$ % saturation vs. Salinity plots to demonstrate the impact of mixing processes. I believe several processes are embedded in the saturation level of $CH_4$: temperature, salinity changes, air-ocean exchanges biogeochemical processes and not only dilution processes by mixing. Also, nowhere in the paper is it made reference to existing $CH_4$ sea ice concentration data in the Antarctic (they are indeed rare, but they do exist). The latter show bulk ice concentration ranges similar to those presented in the water profiles of this paper, which make it very difficult to explain 22% reduction of $CH_4$ saturation in the whole (25-50m thick) mixed layer from melting contribution of 1m thick sea ice, even when using minimal sea ice values. Plotting $CH_4$ concentration values vs. Salinity does not show any trend from West to East along the transect, probably due to partial re-equilibration with the atmosphere (on this I agree with the authors).

The authors also make an overall budget of $CH_4$ air-ocean fluxes for the Ross Sea, based on a 3-days cruise flux measurements… which I think is a bit overstretched. More precaution should be taken in presenting those results (e.g. no clues on what happens in the winter!). The authors already reckon that the unbalance is rather insignificant for their observation period extrapolated to the whole summer period in the Ross Sea.

There are also a certain number of contradictions at several places in the manuscript, which I have listed more completely in the detailed comments here below.

In short, I think the paper is not yet mature for publication in Biogeosciences. My first reaction would be to reject the paper at this stage, but to encourage the authors to rethink the methodology and interpretation of the data and provide us with a new manuscript. Arguments should be presented to dissociate the impact of temperature, salinity, dilution and biogeochemical processes on the observed saturation states, and how these are disentangled from water mass mixing processes.

**Detailed Comments**

l. 15: Delete sentence "Simple box model…waters."…this is more like just a mass balance calculation, from the description in the methods…also there is a contradiction with the methodology where it is stated that advection is considered as negligible!(l. 125)

l. 18: add sentence: "Simple mass balance calculations further suggest that $CH_4$ consumption also contributes to the CH4 undersaturation."

l. 18: Delete "Thus we argue that.." and replace by: "Both $CH_4$ consumption and sea ice melting are therefore important drivers of $CH_4$ undersaturation, which implies that the high-latitude area of the Southern Ocean act as a sink for atmospheric $CH_4$ in the summer."

l. 19-20: This is a summer only budget!... cannot be used as an annual budget", not knowing what happens in the winter!..

l.47-48: "storage of $CH_4$ in ice crystals is usually a minor source compared to sea ice brines (bubbles or dissolved) and under-ice sea water in the Arctic (..)."

l.53: "we propose that (i)…"

l. 54: "..our results from a summer West-East field transect in the Ross Sea…"

l. 54: "and (ii) sea ice melting.."

l. 47-54: This section should refer to the work of Damm et al. on in-situ sea ice CH4 production and impact on air-sea fluxes in the Arctic

l. 56: "Hydrographic measurements and.."

l. 57: I am missing here the description of the actual measurements!.. CTD, precisions…actually, move material from section 2.2

l. 57-71: This is not "Method"!.. but more general statements about ocean circulation in the Ross Sea.. This should be moved to "Introduction" (after lines 32-34) , or eventually used in the discussion section

l. 57-58: Figure 2 and Tables S1 are "Results", not methods

l. 76: Table S2 ans S3 are Results, not Methods

l. 76-77: not clear what the "total volume" is in this case..

l. 78: "$CH_4$ sampling and analysis"

l. 79-80: "onboard…January 2020".. move to section 2.1

l. 85-87: Move to section 2.1

l. 91: "is calculated by"

l. 95: "-1.9 to 1.4°C"

l. 95: "sampling with the atmospheric $CH_4$…"

l. 96: South Pole data.. why not use data from closer by?...Arrival Heights?..([ftp://ftp.niwa.co.nz/CH4/arch](ftp://ftp.niwa.co.nz/CH4/arch))

l. 101: "2.3 Flux Density Calculations"

l. 107-110: Wind data: I am sure there were true wind data recordings on the Xuelong 2, isn't it?.. It would be more accurate to use those

l. 113: "CH4 budget in the mixed layer": Is this what you also refer to as "box model" elsewhere in the paper.should be mentioned!.. is it really a "box model"?.. it is indeed more of a mass balance calculation than a "box model"

l. 114: "In order to give a first estimate of…"

l. 123: This is a true equation for Fvd..should have the status of an equation on a separate line

l. 124-125: "advection… negligible".. in contradiction with what is said in the abstract!

l. 127: "sink for the mixed layer"

l. 128: Introduce here figures 1 and 2 and supplementary table with $CH_4$ concentration… which should actually be grouped with all the rest of the supplementary material!...

l. 130: Figure 1 d should include a plot of $\Delta\sigma$

l. 131: "temperature gradient" .. not differences, given the units..

l. 141: Introduce also here a few words on the CH4 concentrations, referring to the table, and indicating low variability overall (1.5-5.2 nM). Figure 1b should also show the CH4 concentration profiles. Actually Figure 1b deserves to be an isolated figure.

l. 150: "(Figure 2b, Table S2)

l. 155: "most heterogeneous region"

l. 156: "was found only closer to the ice sheet and at depth (Table S2)"

l. 161: "in February 2020"

l. 164: " (Figure 3d, stations R1-R5…"

l. 166-172: This is already "Discussion"

l. 163-164: However, $CH_4$ concentrations are relatively constant in surface waters throughout (ca. 3 nM) so this trend could be S/T impact on saturation rather than dilution through mixing with melted sea ice.

l. 170-172: "We found…".. Where is the blue arrow in Figure 3e coming from?.. If I understood correctly, you are using the numbers from Figure 3d, correct?...But there, the change in saturation level could also result from the combination of other processes such as the effect of temperature and salinity on solubility (which is used to calculate the saturation) or exchanges with the atmosphere, correct?..

In fact I am a bit disturbed by the use of a % saturation vs. salinity plot to discuss mixing processes, while obviously temperature and salinity changes should also affect the saturation numbers. Why not use simply a $CH_4$ concentration vs. salinity plot, as shown here below (built from the values in the supplementary table, in the same way you built your Figure 3e)?

[Figure]

There you see a similar arrangement than in your figure 3e, but clearly the concentration of the mixed layer waters (same criteria as yours – all 50 m depth in this case), shows no trend with salinity... dynamics of CH$_4$ does not indicate dilution from melting sea ice... the CH$_4$ bulk concentration of which could be close to SW (and not negligible!).. see further comments..

l. 173: "Contribution of Production/oxidation from mass balance calculations"... this is not really a box model, and you are talking about those biogeochemical processes in this section

l. 187: "Figure 3e" .. there is no MSW in Figure 3e

l. 189: "in a negative CH$_4$ gradient.." a gradient is a slope.. I don't think you are tracking changes of slopes

l. 190: Figure S2 is tricky to interpret.. it is strange that linear correlations come out while 4 end members are involved (?).. Is it that mixing of AASW is mainly with SW, and not CDW (which would inverse the trend)

l. 191: "Hence, superimposed to the effect of microbial oxidation of CH$_4$ (section 3.4)..."

l. 192-194: Why supersaturation, then?...and why limited to the stations R1 to R4?.. Clearly the distribution of supersaturation in Figure 1b (top 100m) "roots" into the coastal areas.. could there be enrichment from shallow sedimentary sources (e.g. triggered by tidal forcing?) Another source might be ISW (inContact with sediments upstream), but there doesn't seem to be higher CH4 at those depths in your profiles..

l. 196: "caused by dilution"... but, as mentioned before, undersaturation could also result from T/S changes, where is the balance with dilution?

l. 202-205: same comment as above! You should estimate the contribution of each process to demonstrate that dilution is indeed the main factor!... Looking at the figure above, there is no trend in surface waters with the decrease of salinity... but of course, there exchange with air might have blurred the signature…

l. 211: "box model" … rather "Mass Balance calculation"?

l. 212: "increased its contribution in the east" …isn't this in contradiction with Figure 3d?

l. 215: "In the west…"  not east

l. 216: "injected freshwater"…If I understood correctly from previous saying in the paper, this refers to sea ice melt, correct?.. I am a bit surprised that it would have affected the whole mixed layer: sea ice is maximum 1m thick, the mixed layer is 25-50m. Sea ice $CH_4$ concentrations in the Antarctic are not very well documented, but they do exist. Jacques et al., 2021 report on a range of 1.5 to 7.4 nM for bulk $CH_4$ concentration in McMurdo Sound, with a mean around 3 nM. This value is similar to the values for waters in this paper (1.7 to 5 nM). I am therefore not convinced of the impact of dilution from melting sea ice on mixed layers concentrations. Even supposing a minimum sea ice concentration of 1 nM, it would be a factor 3 to 5 lower than SW values, while the mixed layer is a factor 25-50 thicker!…again, plotting $CH_4$ concentration vs. salinity does not show any trend..

l. 218: "exposition to the air"

l. 218: "30 days ice free period in the east"... but this is where Fig. 3d shows minimum supersaturation, correct?.. contradiction?

l. 222: "suggests".. why not only continuous dilution?

l. 223: "previous study" references?

l. 229: "the compared results among stations"... I don't understand. Please rephrase!...

l. 231: rather "Summer CH4 uptake in the Ross Sea"

l. 255-256: you should also insist on the fact that no CH4 data is available today, which hampers the possibility of providing an annual flux budget!..

l. 260: "sea ice melting is likely to enhance..".. this is actually not demonstrated by the CH4 data in this paper..

l. 265: "which underlines the potential significance.."

**Suppl. Table CH4:** please use contrasting background to better define the stations data

**Figure 1:** should show ISW out of the Ross Ice Shelf

**Figure 1b**, should be a separate figure

**Figure 3 caption:**

a) define "surface"
b) Heeschen et al. do not give CDW values, if I remember correctly, but WSDW with a minimum of 0.4 nM $CH_4$

**Figure S3:** where is this scheme commented in the text?.. Maybe I missed it!..

---

## Referee Comment (RC2)

Review of:
**Significant methane undersaturation during austral summer in the Ross Sea (Southern Ocean)**
Ye et al.

**General comments:**

In this manuscript, the authors present an interesting data set of CH4 concentration profiles showing the distribution of CH4 in the water column at 10 locations in the Ross Sea. The measurements were carried out during the austral summer in January 2020 on board of the R/V Xuelong 2. Methane concentration data in the Southern Ocean is scarce, and in this sense, the data set represents a relevant contribution to the community. Nevertheless, the manuscript requires modifications before is suitable for publication.

In terms of the structure of the manuscript, I missed a robust and organized section describing the data collection and methods to support subsequent results and discussion. It is my impression, that the text does not follow a coherent line of thought, jumping back and forth between topics and figures, which makes it hard to trace exactly how the authors arrived to the given results and conclusions. Please see specific comments below.

As I mentioned before, the data presented in the manuscript is interesting. However, a deep analysis of the different processes involved in the CH4 dynamics is missing. The authors focused much of the attention to the dilution effects due to sea-ice melting, leaving aside other relevant processes. I would suggest following a process-based analysis where the relative importance of each mechanisms is assessed, instead of having the melting sea-ice as a central line.

Following are my comments, which I hope can contribute to the improvement of this manuscript.

**Specific comments:**

Abstract:

Is "fresh water injection" considered to be the same process as "advection"?  In L.124 is stated that advection is negligible. Please clarify.

L.19-L.21 Please remove the sentence starting with "We estimated that the Southern Ocean…".  This result can be used as part of the discussion to give some perspective to the potential relevance of the region in the global context. However, using three days of data from one specific region to make a final statement about the role of the whole Southern Ocean as a sink or source of CH4 is not appropriate.

Introduction:

L.30 I guess "emissions" refer to the "net global oceanic emissions".

L.53 remove "on the basis of our results"

L.54 CH4 consumption is not mentioned here, while in the abstract is stated to be as equally important as the sea-ice melting. Please clarify and follow a consistent rhetoric throughout the manuscript.

A paragraph describing the main processes associated with the CH4 cycle would be very useful to contextualize the discussion and to aid non-expert readers. This could be included after L.31 and could be expressed, for example, using the terms of eq.3 (air-sea flux, diffusion, advection,

production/oxidation, etc.), including the relevant aspects of surface CH4 and water-column distribution.

Materials and Methods:

L.55 I suggest renaming this section to "Data and Methods".

After L.55 start by describing the study site and measurements.

L.56 re-name this section as "Hydrographic data and water mass classification"

L.57-L.58 looks more like results (including Fig.2).

L.58 the definition of the different water masses as described in the literature (including Table S1) should be moved further down in the methods section.

L.59-L.71 is not methods. Should be moved to introduction.

L.72-L.77 this paragraph should be moved further down. Please first describe the site and measurements, before addressing how the data was analyzed.

L.74 what do you mean by sectional area? Aren't the measurements taken at individual locations each time? Please clarify.

L.78 please rename. There are two sub-sections named "sampling and analysis". Make sure an adequate name is given to each section and sub-section.

L.80-L.81 refer to Fig.1 after the sentence "The CH4 distribution was measured…". Also please specify at which nine stations where the samples taken.

L.82 even if a detailed description of the sampling method is given in Zhan et al. please include relevant information here, such as sampling depths, measurement times, etc.

L.85 move the sentence "Hydrographic data were collected…" to the section "Hydrographic data and water mass classification", which should follow this section.

L.101 please re-name (see previous comment for L.78)

L.108 why use an average wind speed for the gas transfer velocity? This might introduce significant biases in the flux calculations. If wind data are not available during the research cruise (which I would find strange), there are other resources with sufficient resolution that could be used for the analysis. Please reconsider using other alternatives for the k_w calculations or provide the necessary information to support your decision, including discussion of the uncertainty associated to the calculation using the mean wind value.

L.124-L.125 why is it advection considered negligible? This statement contradicts the text in the abstract (L.16) where advection is considered as one of the two mechanisms leading to the depletion of CH4 in surface waters. Please explain.

L.126-L.127 The sentence "Positive values represent…" is confusing. I suggest "Positive values of Fx represent transport of CH4 from the mixed layer to the surroundings, while negative values represent transport into the mixed layer" or similar, if that is what you meant. As I said, it is a bit confusing.

Results:

L.131 here you recognize the relevance of wind as a mixing processes affecting the upper oceanic layer. Why not taking this effect into account when calculating the gas transfer velocity ($k\_w$) and the fluxes? Also, the stratification effects caused by the density changes might be more relevant at low wind speeds. While at higher wind speeds, large part of the flux is most probably driven by wind-induced mixing. This is of course not evident if the mean wind speed is used for the gas transfer velocity calculations.

L.136 Here "Lateral transport of water masses…" is discussed. Is this lateral transport not associated to $CH_4$ advection? How is $CH_4$ advection negligible but advection of water masses relevant? Please explain.

L.139 "Consequently, for the eastern-most stations (R7-R10), the water column…"

L.145-L.147 why is the high concentration in R7 only observed at the bottom and not in the whole water column, even when the "warm" temperature is observed from the surface to the bottom? Can this be $CH_4$ from the sediments in the sea floor? It would be interesting (maybe in the discussion) to briefly explain why this is input from sediments is observed in R7 and not in the other "shallow" stations.

L.148 this section should go together with paragraph in L.130 to L.140, as water masses are also discussed there. Maybe start with a sub-section on water masses, followed with another separate sub-section about $CH_4$ in the water column (i.e. moving L.141-L.147 further down).

L.151 "would it be expected", does that mean that SW (contrary to what was expected) is not trapped in deep troughs? Or it is, actually, trapped?

L.156 "…was found only near the ice sheet (stations R1-R3), where supercooled…"

L.160 the sea-ice data source is only included in the legend of Fig. 3. It should also be included in the methods.

L.163 "…at the Mawson Bank (stations R1 to R5)."

L.163-L.165 refer to Fig.3d

L.170-L.172 it seems here that all other processes involved in the dynamics of $CH_4$ in the mixed layer have already been discarded. I think the sentence "We found that mixing … is responsible for the $CH_4$ undersaturation in the shelf sea" is farfetched at this point of the manuscript. Please present a thorough assessment of the relevant mechanisms involved in the distribution of $CH_4$ or as you call it "budget in the mixed layer" before presenting such strong statement.

L.173 I assume "box model calculation" refers to what is described in Sect. 2.4 as the $CH_4$ budget in the mixed layer. At some point it is also refer to as "mass balance". Please make use of the terminology in a consistent manner throughout the text.

L.175 "…calculated two box models…", please refer to Sect. 2.4, Eq. 3.

L.177 "If we assume that lateral transport of $CH_4$ is zero", why? Please clarify, as in some parts of the manuscript (i.e. the abstract) advection is stated as one of the "important drivers …" while in other parts of the text is described as "negligible".

L.178 what about measurements in station R7?

Discussion:

L.183 "The  of CH4…" use "distribution", instead.

L.186 "Hence, the CH4-poor CDW may play an important role…" why is it then that this mechanisms is not given the same importance as sea-ice melting? To me it seems like the main focus is to highlight sea-ice melting as the cause of CH4 undersaturation, while different mechanisms were also found to be significant for CH4 dynamics. I suggest to not over-focus on one single process as these results are all relevant! Please explore all possibilities.

L.194 the phrase "…may originate from surface water that is sufficiently exposed to the air…" is confusing as, at least during 2020, this western region (stations R1-R5) is the one that was cover with ice for the longest time (Fig. 3). Please explain. Also, could it be the other way around? That this region is most of the time cover by ice and, therefore, with very little interaction with the atmosphere. Thus, CH4 is being stored there due to sediment CH4 production for example (in comparison to the more "open waters" which experience more air-sea exchange). Then again, as I said before, air-sea gas fluxes (and other mechanisms) might also be relevant!

L.195- L.208 I do not think these statements are really supported by your observations. This paragraph is confusing but most of all, it is misleading as the main focus seems to be to justify the relevance of sea-ice melting. I suggest making a detail assessment of the relative importance of each process involved in the distribution of CH4 in the region, and then discuss the role of all the relevant mechanisms.

L.200 "when sea ice melts in the summer, seawater with undersaturated CH4 concentrations then continues to be diluted…which in turns leads to a continuous decrease …within the surface layers" this is, to my understanding, contradicting the previous sentence in L.192 "the oversaturation of CH4…may originate from surface water…".

L.203 "we found that the CH4 saturation was decreased by 22% at the Pennel Bank…compared to that at the Mawson Bank" can you really conclude this from your observations? Why?

L.205-L.208 "As the ice-free areas increase, … due to mixing and …and/or exchange with the atmosphere. Thus, the magnitude of sea-ice melting may determine the degree…" this sounds much more reasonable. The conditions of the ice may actually affect several biogeochemical and physical processes! But not only changes in CH4 due to dilution effects.

L.211 is it really a box model?

L.212 "…were influenced by air-sea exchange (40-70%) in the west…(90%) in the east." These contributions seem relevant, don't they? Again, I do not understand why the speech along the manuscript is around the dilution due to sea ice melting, when other interesting results are also found.

L.213 maybe add some numbers of the relative importance of the vertical diffusion, similar to what is done for air-sea exchange (in percentage, for example).

L.216 is it east or west?

L.216 this "rapid decline in CH4" is not really seen in the west side (if that is what is meant), is it? How? From the data shown here, it seems like the highest saturation values are found in the western side where no decline in CH4 saturation is observed. I also think that in order to reach such a conclusion, measurements capturing the temporal variability of CH4 in each station are necessary, which are not provided here.

L.229 what is it meant with "regulations"?

**Technical corrections:**

L.15 remove "Simple"

L.31 "…the Southern Ocean in the global CH4 cycle"

L.80 "The CH4 vertical distribution…"

L.87 "Triplicate or duplicate CH4 subsamples…"

L.114 remove "roughly"

L.122 in the equation of the Fick's first law, it looks strange to me to express the gradient using subscripts. I would suggest using dC/dh instead of $d_c/d_h$.

L.123 $K_z$ (in italics)

L.135 "stations R9"

L.155 "…heterogeneous region"

L.214 "The sea ice distribution may be responsible…"

L.215 "…melting or incompletely partially melting…"

L.218 "…after completely a complete melting of …"

L.233 "…, which will result in a net take up uptake of CH4…"

L.263 "Our measurements of CH4…"

Throughout the text, refer to figures and tables when introducing and discussing the results.

---

## Author Comment (AC1)

**Response to reviewers: Significant methane undersaturation during austral summer in the Ross Sea (Southern Ocean)**

We would like to express our thanks and appreciation for your interest and comments to our manuscript. Please find below the point-to-point responses to your comments. The original reviewer comments are in black and our responses are coloured blue. Red texts correspond to the revised manuscript without tracked changes.

**General Comments**

This paper presents a nice data set on $CH_4$ water concentration profiles in the Ross Sea during the Summer 2020. This is rare data, which, per se, deserves publication. The data set is interpreted in terms of potential driving processes such as water mass mixing, sea ice melting, biological consumption/production and air-ocean exchange processes. I have however reservations on several aspects of the paper, both in terms of structure of the paper, methodology and conclusions.

*Response: We appreciate your interest in our manuscript. We agree that the manuscript can be further improved.*

My major concern is that the paper argues that $CH_4$ profiles in the mixed layer demonstrate the impact of sea ice melting on the $CH_4$ dynamics in the area. First of all, I have doubts on the pertinent use of $CH_4$ % saturation vs. Salinity plots to demonstrate the impact of mixing processes. I believe several processes are embedded in the saturation level of $CH_4$: temperature, salinity changes, air-ocean exchanges biogeochemical processes and not only dilution processes by mixing.

*Response: We agree with the reviewer that $CH_4$ saturation was influenced by several aspects. The saturation was intended to be used for convenient comparison with the equilibrium concentration. Now we have changed the saturation to concentration when discussing the impact of mixing processes.*

Also, nowhere in the paper is it made reference to existing $CH_4$ sea ice concentration data in the Antarctic (they are indeed rare, but they do exist). The latter show bulk ice concentration ranges similar to those presented in the water profiles of this paper, which make it very difficult to explain 22% reduction of $CH_4$ saturation in the whole (25-50m thick) mixed layer from melting contribution of 1m thick sea ice, even when using minimal sea ice values. Plotting $CH_4$ concentration values vs. Salinity does not show any trend from West to East along the transect, probably due to partial re-equilibration with the atmosphere (on this I agree with the authors).

*Response: Thank you for the suggestion. We have cited the Jacques et al. (2021) for the bulk $CH_4$ concentration in sea ice. We agree with the reviewer that it is difficult to separate the impact of sea ice melting on $CH_4$ saturation from other processes (e.g., water mixing, microbial oxidation). We have re-evaluated the role of sea ice melting in*

*CH₄ undersaturation and found that is far more complicated than we thought before. The sea ice may a source for the bulk CH₄ in ambient seawater but sea ice melting may facilitate the local water mixing and enhanced the vertical convention, which eventually lowers the surface CH₄ concentration. Since we have no sea-ice data, we decided to deleted the relevant speculations (sea ice melting) in the text. We have re-written the Discussion, discussed the role of water mixing, oxidation, and gas exchange in CH₄ undersaturation.*

The authors also make an overall budget of CH₄ air-ocean fluxes for the Ross Sea, based on a 3-days cruise flux measurements… which I think is a bit overstretched. More precaution should be taken in presenting those results (e.g. no clues on what happens in the winter!). The authors already reckon that the unbalance is rather insignifint for their observation period extrapolated to the whole summer period in the Ross Sea

*Response: We agree. Winter data is important in understanding how sea ice works on CH₄ dynamics. In the revised manuscript, we have highlighted that our calculated fluxes are certainly an underestimation since CH₄ measurements in the Ross Sea as well as in the Southern Ocean from other seasons are largely missing. Besides, the discussion of CH₄ flux are more cautious and only constrained in summer. We realize that the size of dataset is too small to give an overall estimation of CH₄ flux in the Ross Sea but we believe that the conclusions of the study are adequate for the dataset. Importantly, one purpose of this study is calling upon future work in the region, or even in the Southern Ocean, since it is the difficulty to access what makes it so difficult to obtain larger spatial and temporal coverage.*

There are also a certain number of contradictions at several places in the manuscript, which I have listed more completely in the detailed comments here below. In short, I think the paper is not yet mature for publication in Biogeosciences. My first reaction would be to reject the paper at this stage, but to encourage the authors to rethink the methodology and interpretation of the data and provide us with a new manuscript. Arguments should be presented to dissociate the impact of temperature, salinity, dilution and biogeochemical processes on the observed saturation states, and how these are disentangled from water mass mixing processes.

*Response: We thank the reviewer for pointing out these confusions and encouraging us to revise. We agree that the clarity of the manuscript can be further improved. Below are our point-to-point responses to the concerns.*

**Detailed Comments**

l. 15: Delete sentence "Simple box model…waters."…this is more like just a mass balance calculation, from the description in the methods…also there is a contradiction with the methodology where it is stated that advection is considered as negligible!(l. 125)

*Response: Done. Sorry for the misleading statement. The word "advection" in this sentence has been deleted and the role of advection in CH₄ distribution has been discussed in "Section 4.2 CH₄ dynamics in the surface mixed layer".*

l. 18: add sentence: "Simple mass balance calculations further suggest that CH4 consumption also contributes to the CH4 undersaturation."
*Response: Thank you. The sentence has been changed as "the $CH_4$ concentrations in the mixed layer were mainly driven by air-sea exchange and diapycnal diffusion between surface and subsurface layer, as well as in-situ consumption of $CH_4$".*

l. 18: Delete "Thus we argue that.." and replace by: "Both CH4 consumption and sea ice melting are therefore important drivers of CH4 undersaturation, which implies that the high-latitude area of the Southern Ocean act as a sink for atmospheric CH4 in the summer."
*Response: This sentence has been deleted.*

l. 19-20: This is a summer only budget!... cannot be used as an annual budget", not knowing what happens in the winter!..
*Response: We agree. The sentence has been modified as "the Ross Sea was a significant sink for atmospheric $CH_4$ during the austral summer".*

l.47-48: "storage of CH4 in ice crystals is usually a minor source compared to sea ice brines (bubbles or dissolved) and under-ice sea water in the Arctic (..)."
*Response: Thank you. Done.*

l.53: "we propose that (i)…"
*Response: This sentence has been modified as "the objectives of our study were (i) to determine the distribution of $CH_4$ in the water column of the Ross Sea, (ii) to decipher the major processes affecting the $CH_4$ water column distribution and (iii) to determine the role of the Ross Sea as a source or sink of atmospheric $CH_4$".*

l. 54: "..our results from a summer West-East field transect in the Ross Sea…"
*Response: This sentence has been deleted.*

l. 54: "and (ii) sea ice melting.."
*Response: Done.*

l. 47-54: This section should refer to the work of Damm et al. on in-situ sea ice CH4 production and impact on air-sea fluxes in the Arctic
*Response: Thank you for the suggestion. The work of Damm et al. (2015; 2018) have been referred in the text.*

l. 56: "Hydrographic measurements and.."
*Response: Changed to "Study site and hydrographic measurements".*

l. 57: I am missing here the description of the actual measurements!.. CTD, precisions…actually, move material from section 2.2

*Response: Done.*

l. 57-71: This is not "Method"!.. but more general statements about ocean circulation in the Ross Sea.. This should be moved to "Introduction" (after lines 32-34) , or eventually used in the discussion section
*Response: Have been moved to the* "Introduction".

l. 57-58: Figure 2 and Tables S1 are "Results", not methods
*Response: Have been moved to the "Results".*

l. 76: Table S2 and S3 are Results, not Methods
*Response: Have been moved to the "Results".*

l. 76-77: not clear what the "total volume" is in this case..
*Response: It is the volume of all water masses at a specific station, determine by the water depth.*

l. 78: "CH4 sampling and analysis"
*Response: Done.*

l. 79-80: "onboard…January 2020".. move to section 2.1
*Response: Done.*

l. 85-87: Move to section 2.1
*Response: Done.*

l. 91: "is calculated by"
*Response: Done.*

l. 95: "-1.9 to 1.4°C"
*Response: Done.*

l. 95: "sampling with the atmospheric CH4…"
*Response: Done.*

l. 96: South Pole data.. why not use data from closer by?...Arrival Heights?..(ftp://ftp.niwa.co.nz/CH4/arch)
*Response: Thank you for the suggestion. We believe the data suggested by the reviewer is not necessarily appropriate for this study since it contains integrated values over 120 m above sea level.*

l. 101: "2.3 Flux Density Calculations"
*Response: Thank you. Done.*

l. 107-110: Wind data: I am sure there were true wind data recordings on the Xuelong 2, isn't it?.. It would be more accurate to use those

*Response: Thank you for the suggestion, the ship-measured wind data has been used in the revised manuscript.*

l. 113: "CH4 budget in the mixed layer": Is this what you also refer to as "box model" elsewhere in the paper. should be mentioned!.. is it really a "box model"?.. it is indeed more of a mass balance calculation than a "box model"

*Response: This sentence has been changed to "CH$_4$ mass balance calculation in the mixed layer".*

l. 114: "In order to give a first estimate of…"

*Response: Thank you. Done.*

l. 123: This is a true equation for Fvd..should have the status of an equation on a separate line

*Response: Yes, we agree. Done.*

l. 124-125: "advection… negligible".. in contradiction with what is said in the abstract!

*Response: Sorry for the misleading statement. The abstract has been modified to avoid contradiction.*

l. 127: "sink for the mixed layer"

*Response: This sentence has been modified as "Positive values of F$_x$ indicate a production of CH$_4$ in the mixed layer, while negative values indicate a consumption of CH$_4$ in the mixed layer".*

l. 128: Introduce here figures 1 and 2 and supplementary table with CH4 concentration… which should actually be grouped with all the rest of the supplementary material!...

*Response: Thank you for the suggestion. The supplement has been re-grouped as suggested. Please note that Figure 1a has been separated and as a single figure in the revised text.*

l. 130: Figure 1 d should include a plot of $\Delta\sigma$

*Response: The density plot has been added (Figure 2c in the revised version).*

l. 131: "temperature gradient" .. not differences, given the units..

*Response: Done.*

l. 141: Introduce also here a few words on the CH4 concentrations, referring to the table, and indicating low variability overall (1.5-5.2 nM). Figure 1b should also show the CH4 concentration profiles. Actually Figure 1b deserves to be an isolated figure.

*Response: Agree. The sentences have been modified. We also separated Figure 1b as a*

*new figure. The profile of $CH_4$ concentration has been added in Figure 2d.*

l. 150: "(Figure 2b, Table S2)
*Response: Done.*

l. 155: "most heterogeneous region"
*Response: Done.*

l. 156: "was found only closer to the ice sheet and at depth (Table S2)"
*Response: Done.*

l. 161: "in February 2020"
*Response: Done.*

l. 164: " (Figure 3d, stations R1-R5…"
*Response: Done.*

l. 166-172: This is already "Discussion"
*Response: Thank you for the suggestion. These sentences have been moved to the "Discussion".*

l. 163-164: However, CH4 concentrations are relatively constant in surface waters throughout (ca. 3 nM) so this trend could be S/T impact on saturation rather than dilution through mixing with melted sea ice.
*Response: Thank you for the suggestion. We agree that the S/T can influence $CH_4$ saturation. However, in our cases, this impact is limited because the changes in S (33.7-34.8) and T (-1.8-1.4 °C) are small. Please see the Table below. We used the highest and lowest T and S, with a constant $CH_4$ concentration, to calculate the saturation that yields a maximum difference of 9% (which is mainly caused by the T differences). In contrast, 1.1 unit differences of salinity yield only 1% difference in saturation. In our study, the mean temperature at the surface (<5 m) in the west and east were 0.1 vs. -0.1 °C and therefore the impact of temperature on $CH_4$ saturation is limited. Instead, the differences in $CH_4$ concentrations (3.4 nM in the west vs. 2.8 nM in the east) were mainly caused by convection, water mixing (with 0.2-unit differences in salinity), and air-sea gas exchange. We have now indicated both concentration and saturation in the text.*

| $CH_4$ conc. | T | S | Saturation* |
|---|---|---|---|
| 3.4 nM | -1.8 | 34.8 | 90% |
| | 1.4 | 34.8 | 98% |
| | 1.4 | 33.7 | 97% |
| | -1.8 | 33.7 | 89% |
| 2.8 nM | -1.8 | 33.7 | 73% |

*Calculated from Wiesenburg and Guinasso (1979)

l. 170-172: "We found…".. Where is the blue arrow in Figure 3e coming from?.. If I understood correctly, you are using the numbers from Figure 3d, correct?...But there, the change in saturation level could also result from the combination of other processes such as the effect of temperature and salinity on solubility (which is used to calculate the saturation) or exchanges with the atmosphere, correct?..

In fact I am a bit disturbed by the use of a % saturation vs. salinity plot to discuss mixing processes, while obviously temperature and salinity changes should also affect the saturation numbers. Why not use simply a CH4 concentration vs. salinity plot, as shown here below (built from the values in the supplementary table, in the same way you built your Figure 3e)?

[Figure]

There you see a similar arrangement than in your figure 3e, but clearly the concentration of the mixed layer waters (same criteria as yours – all 50 m depth in this case), shows no trend with salinity… dynamics of CH4 does not indicate dilution from melting sea ice… the CH4 bulk concentration of which could be close to SW (and not negligible!).. see further comments..

*Response: Yes, there are multiple processes that affect CH₄ concentration and saturation. Thank you for your suggestion, we have re-built the Figure 3e (in the revised manuscript it is Figure 5) by using concentration rather than saturation. We have been also re-written the discussion part with regard to this figure.*

[Figure]

Figure 5: Mixing diagram between salinity and CH$_4$ concentration on the Ross Sea shelf. Black crosses are all samples and blue stars are samples from the surface layer. Arrows represent controlling processes (green for air-sea exchange, orange for microbial oxidation, and red for water mixing). The green dashed line represents CH$_4$ in equilibrium with respect to the atmosphere at a constant temperature (mean of all samples: -1.4 °C). The open circle represents the endpoint of AASW that was obtained from station R9 with the lowest salinity; the plus represents the endpoint of sediment-influenced water that was obtained from the near-bottom water (200 m) at station R7; the diamond represents the endpoint of SW (highest salinity) that was obtained from the bottom water (1018 m) at station R3; the triangle represents the endpoint of WSDW (Weddell Sea Deep Water) that was obtained from *Heeschen et al.* (2004). The purple shading indicates the mixing results for MCDW and MSW.

l. 173: "Contribution of Production/oxidation from mass balance calculations"... this is not really a box model, and you are talking about those biogeochemical processes in this section
*Response: Agree. This head title has been changed as "Contribution of advection and production/oxidation from mass balance calculations".*

l. 187: "Figure 3e" .. there is no MSW in Figure 3e
*Response: The word "MSW" has been changed to "SW and AASW". The MSW is also modified from the mixing between SW and CDW.*

l. 189: "in a negative CH4 gradient.." a gradient is a slope.. I don't think you are tracking changes of slopes
*Response: Thank you. This sentence has been deleted.*

l. 190: Figure S2 is tricky to interpret.. it is strange that linear correlations come out while 4 end members are involved (?).. Is it that mixing of AASW is mainly with SW,

and not CDW (which would inverse the trend)

*Response: The AASW is also mixed with MCDW in shallow water depth. This figure has been deleted.*

l. 191: "Hence, superimposed to the effect of microbial oxidation of CH4 (section 3.4)…"

*Response: Thanks, done.*

l. 192-194: Why supersaturation, then?...and why limited to the stations R1 to R4?.. Clearly the distribution of supersaturation in Figure 1b (top 100m) "roots" into the coastal areas.. could there be enrichment from shallow sedimentary sources (e.g. triggered by tidal forcing?) Another source might be ISW (inContact with sediments upstream), but there doesn't seem to be higher CH4 at those depths in your profiles..

*Response: Agree. From limited data, we should say that this question cannot be fully addressed with our data and one of the purposes of this paper is to call upon more observations in the Southern Ocean. Thus, in our paper, we proposed the most probable origin of the high CH$_4$ value, which was attributed to CH$_4$ release from the sediments. Any discussion on the origin of this high CH$_4$ is speculative so we decide to focus the CH$_4$ undersaturation in the main text.*

*Indeed, we have cautiously discussed these CH$_4$ concentration anomalies and did not treat them as outliers because 1) the analytical method is reliable (please see Supporting Information), with a low limit of detection (0.04 nmol/L, Table S4) and good precision (5%) and accuracy (0.3 nmol/L) (Table S5). Together with the cautious sampling, storage, and transport of seawater, we believe that the CH$_4$ data are highly reliable and reflected the field situation; 2) those values have been double-checked and analysed at least 2–3 times; 3) We cannot exclude the possibilities of the presence of other CH$_4$ sources in the Ross Sea. Thank you for your suggestion. This part has been modified and further speculation has been deleted to keep the manuscript simple and concise.*

l. 196: "caused by dilution"... but, as mentioned before, undersaturation could also result from T/S changes, where is the balance with dilution?

*Response: Please see our response above. The concentrations were different between ice-free and ice-covered areas. We have modified the expressions in the text. In addition, since we have no sea ice data, the revised manuscript is highlight in the observation of CH$_4$ undersaturation but light in the role of sea ice melting in CH$_4$ distributio.*

"Surface CH$_4$ concentrations ranged from 2.3 to 3.5 nM and are at the lower end of the range of CH$_4$ concentrations (1.5-7.4 nM) reported for bulk CH$_4$ concentration in sea ice in the McMurdo Sound (*Jacques et al.*, 2021). It is reasonable to assume that freshwater from melting ice mixes with the ambient seawater. Provided that CH$_4$ consumption (i.e. $F_x$) was constant at all stations the resulting CH$_4$ flux by advection ($F_{ad}$) is in the range from 0.05 to 1.26 μmol m$^{-2}$ day$^{-1}$ (except for station R5 where the most pronounced air-sea exchange occurred), indicating that $F_{ad}$ is a source for CH$_4$ in the mixed layer. The calculated $F_{ad}$ was negatively correlated with the ice conditions

(r$^2$=0.67, $p$<0.05, Figure 6b), which implies a free-ice conditions may enhance the advection of CH$_4$-enriched water while the ice-covered conditions may favour lower transport of CH$_4$-enriched waters to the mixed layer."

l. 202-205: same comment as above! You should estimate the contribution of each process to demonstrate that dilution is indeed the main factor!... Looking at the figure above, there is no trend in surface waters with the decrease of salinity... but of course, there exchange with air might have blurred the signature…

*Response: Estimation of the contribution of each process to CH$_4$ undersaturation is difficult based on limited data at current stage. However, we have tried discuss the effect of mixing and did not imply it was the main driver. Please see the "section 4.2 CH$_4$ dynamics in the surface mixed layer" in the revised version.*

"we found that in the west, the uptake by air-sea exchange accounts for about 59% of the CH$_4$ while CH$_4$ oxidation was the main process (96%) to remove CH$_4$ from the surface mixed layer. In the east, the downward diffusion of CH$_4$ accounts for 84% of the CH$_4$ removal from the surface mixed layer, becoming the most important process to maintain the status of surface CH$_4$ undersaturation."

l. 211: "box model" ... rather "Mass Balance calculation"?
*Response: Have been changed.*

l. 212: "increased its contribution in the east" …isn't this in contradiction with Figure 3d?
*Response: Sorry for the unclear expression. We mean the contribution of air-sea exchange accounts for 40-70% in the west and 90% in the east that was compared with the contribution of vertical diffusion. This sentence has been deleted. Please also note that it is the low CH$_4$ concentration at the air-sea interface that determines the strong air-sea flux, not inverse. Indeed, the F$_{air}$ was stronger in the east due to free of sea ice. However, other processes, such as dilution via water mixing between AASW and SW (strong in the east, Figure 2b), vertical diffusion into deeper waters and microbial oxidation offsets the influence of air-sea exchange on CH$_4$ concentrations, resulting a relatively low CH$_4$ in the east. This, in turn, explains the influence of sea ice melting on CH$_4$ uptake: more sea ice melting-enhanced water mixing-lower CH$_4$ concentration at the surface-stronger gas exchange-more CH$_4$ uptake from the air.*

l. 215: "In the west…" not east
*Response: Sorry for the typo. Has been modified.*

l. 216: "injected freshwater"…If I understood correctly from previous saying in the paper, this refers to sea ice melt, correct?.. I am a bit surprised that it would have affected the whole mixed layer: sea ice is maximum 1m thick, the mixed layer is 25-50m. Sea ice CH4 concentrations in the Antarctic are not very well documented, but they do exist. Jacques et al., 2021 report on a range of 1.5 to 7.4 nM for bulk CH4 concentration in McMurdo Sound, with a mean around 3 nM. This value is similar to

the values for waters in this paper (1.7 to 5 nM). I am therefore not convinced of the impact of dilution from melting sea ice on mixed layers concentrations. Even supposing a minimum sea ice concentration of 1 nM, it would be a factor 3 to 5 lower than SW values, while the mixed layer is a factor 25-50 thicker!…again, plotting CH4 concentration vs. salinity does not show any trend..

*Response: Thank you for your suggestion. The reference (Jacques et al., 2021) has been cited in the text. We agree that the impact of dilution from melting sea ice on CH₄ is difficult to track in this study, so we have rephrased the words and sentences in the revised version (the sea ice melting part has been deleted.). The specific mixed layer depths have been given in the Table S3, which demonstrates the mixed layer depth ranged from 7 m to 39 m.*

l. 218: "exposition to the air"
*Response: This sentence has been deleted.*

l. 218: "30 days ice free period in the east"... but this is where Fig. 3d shows minimum supersaturation, correct?.. contradiction?
*Response: Please see the response to Line 212. Air-sea exchange is not the only factor that influence the CH₄ concentration in the mixed layer. Instead, strong water mixing with AASW in the mixed layer (compared with that in the west) and vertical convection together caused a low CH₄ concentration at the surface, which in turn, leads to a strong CH₄ uptake from the air. Longer timescale of absorption of CH₄ from the air did not equal to higher CH₄ concentration in the mixed layer because the presence of CH₄ removal. Please see the revised manuscript. However, we believe this is speculative and has been deleted.*

l. 222: "suggests".. why not only continuous dilution?
*Response: These sentences has been deleted.*

l. 223: "previous study" references?
*Response: Heeschen et al. (2004) has been added in the text.*

l. 229: "the compared results among stations"... I don't understand. Please rephrase!...
*Response: Sorry for the unclear expression. The sentence has been rephrased as "in the west, CH₄ oxidation was the main process (96%) to remove CH₄ from the surface mixed layer".*

l. 231: rather "Summer CH4 uptake in the Ross Sea"
*Response: We have highlighted that the CH₄ flux were only calculated in summer and emphasized the importance of CH₄ data from other seasons.*

l. 255-256: you should also insist on the fact that no CH4 data is available today, which hampers the possibility of providing an annual flux budget!..
*Response: Thank you for the suggestion. Done.*

l. 260: "sea ice melting is likely to enhance.."... this is actually not demonstrated by the CH4 data in this paper..

*Response: Agree. The sea ice melting is far more complicated than we thought before. The sea ice may a source to dissolve CH$_4$ but the sea ice melting may facilitate the local water mixing (e.g., increase in AASW from west to east, see section 3.1, Figure 2b), increase the vertical convention of CH$_4$, and increase the timescale of surface area exposition to the air, which eventually leads more uptake of atmospheric CH$_4$. This part has been deleted because it can-not be solved at present by our data alone.*

l. 265: "which underlines the potential significance.."
*Response: Thank you. Done.*

**Suppl. Table CH$_4$:** please use contrasting background to better define the stations data
*Response: Thank you for the suggestion. Done.*

**Figure 1:** should show ISW out of the Ross Ice Shelf
*Response: Please notice the green line that shows the ISW out of the Ross Ice Shelf.*

**Figure 1b**, should be a separate figure
*Response: Done.*

**Figure 3 caption:**
a) define "surface"
b) Heeschen et al. do not give CDW values, if I remember correctly, but WSDW with a minimum of 0.4 nM CH$_4$
*Response: Thank you for the suggestion. The Figure has been modified, please see above response (Line 170-172).*

**Figure S3:** where is this scheme commented in the text?.. Maybe I missed it!..
*Response: At the end of "Discussion". It is an assumed mechanism that shows how CH$_4$-undersaturated water forms during different seasons. We do not have winter data so at this stage it may a speculation to some extent. It is our purpose that appeal more attentions on CH$_4$ dynamics in the Southern Ocean to deeply understand the mechanism.*

---

## Author Comment (AC2)

**Response to reviewers: Significant methane undersaturation during austral summer in the Ross Sea (Southern Ocean)**

We thank the reviewer for the thorough review and constructive suggestions. Please see the point-to-point responses below. The original reviewer comments are in black and our responses are coloured blue. Red texts correspond to the revised manuscript without tracked changes.

**General comments**

In this manuscript, the authors present an interesting data set of CH4 concentration profiles showing the distribution of CH4 in the water column at 10 locations in the Ross Sea. The measurements were carried out during the austral summer in January 2020 on board of the R/V Xuelong 2. Methane concentration data in the Southern Ocean is scarce, and in this sense, the data set represents a relevant contribution to the community. Nevertheless, the manuscript requires modifications before is suitable for publication.

*Response: Thanks for your interest. We realize that the paper was not clear enough. The paper has been thoroughly modified based on reviewer's comments.*

In terms of the structure of the manuscript, I missed a robust and organized section describing the data collection and methods to support subsequent results and discussion. It is my impression, that the text does not follow a coherent line of thought, jumping back and forth between topics and figures, which makes it hard to trace exactly how the authors arrived to the given results and conclusions. Please see specific comments below.

*Response: Thank you for the comment. The section "Materials and Methods" has now been re-organized and the description of detailed methods were along with the supplement material (Text S1, Table S4-S5, Figure S1). The revised manuscript is focusing on the $CH_4$ distribution in the Ross Sea and the subsequent consequences of $CH_4$ uptake during summertime.*

As I mentioned before, the data presented in the manuscript is interesting. However, a deep analysis of the different processes involved in the CH4 dynamics is missing. The authors focused much of the attention to the dilution effects due to sea-ice melting, leaving aside other relevant processes. I would suggest following a process-based analysis where the relative importance of each mechanisms is assessed, instead of having the melting sea-ice as a central line. Following are my comments, which I hope can contribute to the improvement of this manuscript.

*Response: We do agree with the reviewer that multi-processes cause the $CH_4$ undersaturation. After re-evaluation of the role sea ice melting in $CH_4$ undersaturation, we found it is far more complicated than we thought before. The main purpose of this*

*study is reporting CH₄ undersaturation in the Ross Sea. Hence, to keep the manuscript simple and concise, the effect of sea ice melting has been deleted in the revised version. In addition, we have now re-written the "Discussion" to separately discuss the impact of water mixing, vertical convection, CH₄ oxidation, and air-sea exchange on CH₄ distribution. The mass balance calculation has been discussed with Figure 6. Please see our detailed response below.*

**Specific comments**
Abstract:
Is "fresh water injection" considered to be the same process as "advection"? In L.124 is stated that advection is negligible. Please clarify.
*Response: Sorry for this unclear expression. The "fresh water injection" refers to the mixing between ice-melting water and ambient seawater. This sentence has been deleted and the abstract has been re-written.*

L.19-L.21 Please remove the sentence starting with "We estimated that the Southern Ocean…". This result can be used as part of the discussion to give some perspective to the potential relevance of the region in the global context. However, using three days of data from one specific region to make a final statement about the role of the whole Southern Ocean as a sink or source of CH4 is not appropriate.
*Response: We agree. This sentence has been deleted.*

Introduction:
L.30 I guess "emissions" refer to the "net global oceanic emissions".
*Response: We have modified as "oceanic CH₄ emissions".*

L.53 remove "on the basis of our results"
*Response: Done.*

L.54 CH4 consumption is not mentioned here, while in the abstract is stated to be as equally important as the sea-ice melting. Please clarify and follow a consistent rhetoric throughout the manuscript.
*Response: Thanks for the suggestion. The sentence has been rewritten in the revised version.*
"the objectives of our study were (i) to determine the distribution of CH₄ in the water column of the Ross Sea, (ii) to decipher the major processes affecting the CH₄ water column distribution and (iii) to determine the role of the Ross Sea as a source or sink of atmospheric CH₄"

A paragraph describing the main processes associated with the CH4 cycle would be very useful to contextualize the discussion and to aid non-expert readers. This could be included after L.31 and could be expressed, for example, using the terms of eq.3 (air-sea flux, diffusion, advection, production/oxidation, etc.), including the relevant aspects of surface CH4 and water-column distribution.

*Response: A paragraph has been added in the "Introduction", please see the revised version.*

Materials and Methods:
L.55 I suggest renaming this section to "Data and Methods".
*Response: Agree. The title has changed as "Methods". The data report in this section has been moved to the section "Results".*

After L.55 start by describing the study site and measurements.
L.56 re-name this section as "Hydrographic data and water mass classification"
*Response: Agree. The section has been re-organized and the title was replaced by "2.1 Study site and hydrographic measurements".*

L.57-L.58 looks more like results (including Fig.2).
L.58 the definition of the different water masses as described in the literature (including Table S1) should be moved further down in the methods section.
*Response: Agree. The sentence (including Fig. 2 and Table S1) has been moved to the "Section 3.1" according to another reviewer's suggestion.*

L.59-L.71 is not methods. Should be moved to introduction.
*Response: Agree. Done.*

L.72-L.77 this paragraph should be moved further down. Please first describe the site and measurements, before addressing how the data was analyzed.
*Response: Done. It was moved in the section "2.5 Water mass abundance calculation".*

L.74 what do you mean by sectional area? Aren't the measurements taken at individual locations each time? Please clarify.
*Response: To calculate the contribution of each water mass, we need to know the total volume of this specific water mass. The measurements were taken as a point, but the calculation on volume needs an "area". So, assuming the water column at each station is a cylinder, then the "point" can represent the "area" and the volume can be calculated by water depth (the "high") and the assuming sectional area (surface).*

L.78 please rename. There are two sub-sections named "sampling and analysis". Make sure an adequate name is given to each section and sub-section.
*Response: Sorry for the typo. Has been corrected as "2.3 Flux density calculations".*

L.80-L.81 refer to Fig.1 after the sentence "The CH4 distribution was measured…". Also please specify at which nine stations where the samples taken.
*Response: Thank you for the suggestion. Done.*
"The samples for the determination of dissolved $CH_4$ concentrations were collected at nine stations (R1-R7 and R9/R10) on a transect along 75°S from 164°E−182°E (Figure 1)"

L.82 even if a detailed description of the sampling method is given in Zhan et al. please include relevant information here, such as sampling depths, measurement times, etc.

*Response: The sampling depth has been mentioned in the section 2.1. The sampling method has introduced after the reference.*

"seawater was transferred to 250 mL borosilicate glass bottles with standard taper stoppers (Corning PYREX®, USA), which were sealed with Apiezon grease (Sigma-Aldrich®, USA) and stored in the dark at 4 °C after 180 µL of saturated $HgCl_2$(aq) solution was added. Samples were analysed immediately after shipment to the home laboratory (storage time < six months)"

L.85 move the sentence "Hydrographic data were collected…" to the section "Hydrographic data and water mass classification", which should follow this section.

*Response: Done.*

L.101 please re-name (see previous comment for L.78)

*Response: Done. Has been titled as "2.3 Flux density calculations".*

L.108 why use an average wind speed for the gas transfer velocity? This might introduce significant biases in the flux calculations. If wind data are not available during the research cruise (which I would find strange), there are other resources with sufficient resolution that could be used for the analysis. Please reconsider using other alternatives for the k_w calculations or provide the necessary information to support your decision, including discussion of the uncertainty associated to the calculation using the mean wind value.

*Response: Thank you for the suggestion. We have now used the ship-measured wind speed to calculate the $k_w$, which yields a mean flux of -0.44 µmol $m^{-2}$ $day^{-1}$. It is well known that air-sea gas exchange is depending on the variability of the wind and on the choice of the wind speeds. Moreover, air-sea gas exchange is mainly influenced by the choice of the model for $k_w$. We did not discuss it in the text because we would like to focus on the topic of $CH_4$ undersaturation in the Ross Sea.*

L.124-L.125 why is it advection considered negligible? This statement contradicts the text in the abstract (L.16) where advection is considered as one of the two mechanisms leading to the depletion of CH4 in surface waters. Please explain.

*Response: Sorry for the misleading. The abstract has been modified to avoid contradiction. As you can see from Figure 3 (in the revised version Figure 4), the study site was surrounded by the Ross Ice Shelf and sea ice during our sampling period. Hence, to simplified the mass balance calculation, we assumed that there was no advection at station R3 (90% ice condition and near the coast) in the surface mixed layer (7-39 m, Table S3 in the revised version). The discussion has been mentioned in the section 4.2.*

L.126-L.127 The sentence "Positive values represent…" is confusing. I suggest

"Positive values of Fx represent transport of CH4 from the mixed layer to the surroundings, while negative values represent transport into the mixed layer" or similar, if that is what you meant. As I said, it is a bit confusing.

*Response: Sorry for the unclear expression. This sentence has been revised.*

"Positive values of $F_x$ indicate a production of CH4 in the mixed layer, while negative values indicate a consumption of CH4 in the mixed layer."

Results:

L.131 here you recognize the relevance of wind as a mixing processes affecting the upper oceanic layer. Why not taking this effect into account when calculating the gas transfer velocity (k_w) and the fluxes? Also, the stratification effects caused by the density changes might be more relevant at low wind speeds. While at higher wind speeds, large part of the flux is most probably driven by wind-induced mixing. This is of course not evident if the mean wind speed is used for the gas transfer velocity calculations.

*Response: Thank you for the suggestion. We have now used the instantaneous wind speed to calculate the CH4 flux. The impact of wind speed on CH4 flux has been discussed in section 4.3 in the revised paper.*

L.136 Here "Lateral transport of water masses…" is discussed. Is this lateral transport not associated to CH4 advection? How is CH4 advection negligible but advection of water masses relevant? Please explain.

*Response: Sorry for the misleading statement. We have now re-built the mass balance calculation followed the reviewer's suggestions. The advection has been now evaluated and discussed in section 4.2.*

L.139 "Consequently, for the eastern-most stations (R7-R10), the water column…"

*Response: Done*

L.145-L.147 why is the high concentration in R7 only observed at the bottom and not in the whole water column, even when the "warm" temperature is observed from the surface to the bottom? Can this be CH4 from the sediments in the sea floor? It would be interesting (maybe in the discussion) to briefly explain why this is input from sediments is observed in R7 and not in the other "shallow" stations.

*Response: Thank you for the comment. From limited data, we should say that this question cannot be fully addressed with our data and one of the purposes of this paper is to call upon more observations in the Southern Ocean. Thus, in our paper, we proposed the most probable origin of the high CH4 value, which was attributed to CH4 release from the sediments. Any discussion on the origin of this high CH4 is speculative so we decide to focus the CH4 undersaturation in the main text.*

*Indeed, we have cautiously discussed these CH4 concentration anomalies and did not treat them as outliers because 1) the analytical method is reliable (please see Supporting Information), with a low limit of detection (0.04 nmol/L, Table S4) and good*

*precision (5%) and accuracy (0.3 nmol/L) (Table S5). Together with the cautious sampling, storage, and transport of seawater, we believe that the CH₄ data are highly reliable and reflected the field situation; 2) those values have been double-checked and analysed at least 2–3 times; 3) We cannot exclude the possibilities of the presence of other CH₄ sources in the Ross Sea.*

L.148 this section should go together with paragraph in L.130 to L.140, as water masses are also discussed there. Maybe start with a sub-section on water masses, followed with another separate subsection about CH4 in the water column (i.e. moving L.141-L.147 further down).
*Response: Thank you for the suggestion. The paragraph has been re-organized*

L.151 "would it be expected", does that mean that SW (contrary to what was expected) is not trapped in deep troughs? Or it is, actually, trapped?
*Response: Sorry for the unclear expression. The sentence has been changed to "SW was trapped in deep troughs (>400 m)", which is related with the calculation of bulk CH₄ concentration in the SW in the section 4.1.*

L.156 "…was found only near the ice sheet (stations R1-R3), where supercooled…"
*Response: Done.*

L.160 the sea-ice data source is only included in the legend of Fig. 3. It should also be included in the methods.
*Response: The sea-ice data source has been described in "Methods".*

L.163 "…at the Mawson Bank (stations R1 to R5)."
*Response: Done. In the revised manuscript, we compared the CH₄ concentrations at most- western (R1−R4) and eastern stations (R7−R10) in discussing the role of sea ice condition in CH₄ uptake.*

L.163-L.165 refer to Fig.3d
*Response: Done. In the revised manuscript Figure 4d.*

L.170-L.172 it seems here that all other processes involved in the dynamics of CH4 in the mixed layer have already been discarded. I think the sentence "We found that mixing … is responsible for the CH4 undersaturation in the shelf sea" is farfetched at this point of the manuscript. Please present a thorough assessment of the relevant mechanisms involved in the distribution of CH4 or as you call it "budget in the mixed layer" before presenting such strong statement.
*Response: Thank you for the constructive suggestion. We have re-organized and re-written the "Discussion". The mechanisms involved in the distribution of CH₄ undersaturation has been separately discussed in the revised version.*

L.173 I assume "box model calculation" refers to what is described in Sect. 2.4 as the

CH4 budget in the mixed layer. At some point it is also refer to as "mass balance". Please make use of the terminology in a consistent manner throughout the text.

*Response: Thank you for the suggestion. We have been used the term "mass balance" in the text.*

L.175 "…calculated two box models…", please refer to Sect. 2.4, Eq. 3.

*Response: Done.*

L.177 "If we assume that lateral transport of CH4 is zero", why? Please clarify, as in some parts of the manuscript (i.e. the abstract) advection is stated as one of the "important drivers …" while in other parts of the text is described as "negligible".

*Response: This is unclear expression. We have added Figure 6 and section 4.2 to discuss the "Surface mixed layer CH4 dynamics". Please see the revised version.*

L.178 what about measurements in station R7?

*Response: In the revised version, the specific mixed layer depths were given in Table S3 and the mass balance calculation for R7 has been added in the text.*

Discussion:

L.183 "The fate of CH4…" use "distribution", instead.

*Response: The "Discussion" has been re-written.*

L.186 "Hence, the CH4-poor CDW may play an important role…" why is it then that this mechanisms is not given the same importance as sea-ice melting? To me it seems like the main focus is to highlight seaice melting as the cause of CH4 undersaturation, while different mechanisms were also found to be significant for CH4 dynamics. I suggest to not over-focus on one single process as these results are all relevant! Please explore all possibilities.

*Response: Thank you for the comment. We have now re-evaluated the involved processes. We agree with the reviewer that water mixing is important in determining the CH4 undersaturation. After the detailed discussion, we found that the water mixing is the primary factor that determines the distribution of CH4 undersaturation in the Ross Sea, superimposed to the prerequisite of CH4 oxidation. Meanwhile, the level of CH4 undersaturation in the subsurface water is the key factor that determines the rate of vertical transfer and how much CH4 could be uptake from the atmosphere during ice-free periods. These have now been discussed and concluded in the text.*

L.194 the phrase "…may originate from surface water that is sufficiently exposed to the air…" is confusing as, at least during 2020, this western region (stations R1-R5) is the one that was cover with ice for the longest time (Fig. 3). Please explain. Also, could it be the other way around? That this region is most of the time cover by ice and, therefore, with very little interaction with the atmosphere. Thus, CH4 is being stored there due to sediment CH4 production for example (in comparison to the more "open waters" which experience more air-sea exchange). Then again, as I said before, air-sea

gas fluxes (and other mechanisms) might also be relevant!

*Response: We agree that the sentence is misleading so that we have deleted it in the revised version. We thank for the reviewer's detailed the discussion that lighting our thoughts. Please noted that the direction of air-sea exchange was up-to-down, which means more $CH_4$ would be added in the surface waters when experience longer ice-free periods. We believe that the air-sea exchange is an important factor that influences the $CH_4$ distribution in surface waters, the interesting thing is that most-eastern stations were more undersaturated with $CH_4$ even they experience more air-sea exchange. This suggest other factors, such as mixing with $CH_4$-poor water, microbial oxidation, and vertical convection, together with gas exchange would determine the degree of $CH_4$ undersaturation. The corresponding discussions have been added in the revised manuscript.*

L.195- L.208 I do not think these statements are really supported by your observations. This paragraph is confusing but most of all, it is misleading as the main focus seems to be to justify the relevance of sea ice melting. I suggest making a detail assessment of the relative importance of each process involved in the distribution of CH4 in the region, and then discuss the role of all the relevant mechanisms.

*Response: Thank you for the comment. We have now re-evaluated these mechanisms and re-written the "Discussion".*

L.200 "when sea ice melts in the summer, seawater with undersaturated CH4 concentrations then continues to be diluted…which in turns leads to a continuous decrease …within the surface layers" this is, to my understanding, contradicting the previous sentence in L.192 "the oversaturation of CH4…may originate from surface water…".

*Response: Sorry for the misleading sentence. We may consider two stages of sea ice melting. (1) Sea ice melting-induced decrease in $CH_4$ may occur at the initiation of melting that enhance the concentration gradient at the ocean-atmosphere interface. (2) No further melt of sea ice (ice free), the surface $CH_4$ either close to the equilibrium concentration with the atmosphere with low gas exchange rates, or keep undersaturated status due to other processes (which is our case). The unclear expressions in Line 192 have been deleted and this part has been re-written.*

"The calculated $F_{ad}$ was negatively correlated with the ice conditions ($r^2$=0.67, $p$<0.05, Figure 6b), which implies a free-ice conditions may enhance the advection of $CH_4$-enriched water while the ice-covered conditions may favour lower transport of $CH_4$-enriched waters to the mixed layer"

L.203 "we found that the CH4 saturation was decreased by 22% at the Pennel Bank…compared to that at the Mawson Bank" can you really conclude this from your observations? Why?

*Response: Since we have no $CH_4$ data in sea ice, we deleted this point.*

L.205-L.208 "As the ice-free areas increase, … due to mixing and …and/or exchange

with the atmosphere. Thus, the magnitude of sea-ice melting may determine the degree…" this sounds much more reasonable. The conditions of the ice may actually affect several biogeochemical and physical processes! But not only changes in CH4 due to dilution effects.

*Response: Thank you for the suggestion.*

L.211 is it really a box model?

*Response: Has been modified as "mass balance calculation".*

L.212 "…were influenced by air-sea exchange (40-70%) in the west…(90%) in the east." These contributions seem relevant, don't they? Again, I do not understand why the speech along the manuscript is around the dilution due to sea ice melting, when other interesting results are also found.

*Response: Sorry for the misleading sentence. The role of sea ice melting has been deleted and other processes have been discussed in the revised version.*

L.213 maybe add some numbers of the relative importance of the vertical diffusion, similar to what is done for air-sea exchange (in percentage, for example).

*Response: Done. Please see the revised manuscript.*

L.216 is it east or west?

*Response: Sorry for the typo, it is west. Has been modified.*

L.216 this "rapid decline in CH4" is not really seen in the west side (if that is what is meant), is it? How? From the data shown here, it seems like the highest saturation values are found in the western side where no decline in CH4 saturation is observed. I also think that in order to reach such a conclusion, measurements capturing the temporal variability of CH4 in each station are necessary, which are not provided here.

*Response: We agree that this sentence may cause ambiguity and therefore it has been deleted.*

L.229 what is it meant with "regulations"?

*Response: Sorry for the unclear expression. This paragraph has been modified.*

Technical corrections:
L.15 remove "Simple"

*Response: Done.*

L.31 "…the Southern Ocean in the global CH4 cycle"

*Response: This sentence has been deleted.*

L.80 "The CH4 vertical distribution…"

*Response: This sentence has been modified as "the determination of dissolved $CH_4$ concentrations…".*

L.87 "Triplicate or duplicate CH4 subsamples…"
*Response: Done.*

L.114 remove "roughly"
*Response: Done.*

L.122 in the equation of the Fick's first law, it looks strange to me to express the gradient using subscripts. I would suggest using dC/dh instead of dc/dh.
*Response: Done.*

L.123 Kz (in italics)
*Response: Done.*

L.135 "stations R9"
*Response: Done.*

L.155 "…heterogeneous region"
*Response: Done.*

L.214 "The sea ice distribution may be responsible…"
*Response: This sentence has been deleted.*

L.215 "…melting or incompletely partially melting…"
*Response: This sentence has been deleted.*

L.218 "…after completely a complete melting of …"
*Response: This sentence has been deleted.*

L.233 "…, which will result in a net take up uptake of CH4…"
*Response: This sentence has been modified.*

L.263 "Our measurements of CH4…"
*Response: Done.*

Throughout the text, refer to figures and tables when introducing and discussing the results.
*Response: Done.*